# On the Self-awareness of Large Reasoning Models' Capability Boundaries

## Abstract

Large Reasoning Models (LRMs) have shown impressive performance on complex reasoning tasks such as mathematics, yet they also display misbehaviors that expose their limitations. In particular, when faced with hard questions, LRMs often engage in unproductive reasoning until context limit, producing wrong answers while wasting substantial computation. This phenomenon reflects a fundamental issue: *current answering paradigms overlook the relationship between questions and LRMs' capability boundaries*. In this paper, we investigate whether LRMs possess *self-awareness of capability boundaries*. We begin by an observation that LRMs may know what they cannot solve through expressed reasoning confidence. For black-box models, we find that reasoning expressions reveal boundary signals, with accelerated growing confidence trajectory for solvable problems but convergent uncertainty trajectory for unsolvable ones. For white-box models, we show that hidden states of the last input token encode boundary information, with solvable and unsolvable problems linearly separable even before reasoning begins. Building on these findings, we propose two simple yet effective optimization strategies: reasoning expression monitoring and hidden states monitoring. Experiments demonstrate that these boundary-aware strategies enable LRMs to avoid unproductive reasoning without sacrificing accuracy, significantly improving reliability and efficiency by cutting token usage up to $62.7 - 93.6\%$. We anonymously open-source at `https://anonymous.4open.science/r/CB-032D/`.

## 1 Introduction

Large Reasoning Models (LRMs) have demonstrated remarkable capabilities on complex reasoning tasks such as mathematics (Guo et al., 2025; Jaech et al., 2024; Ahn et al., 2024). However, they also exhibit a range of misbehaviors that reveal the limitations of their capabilities (Kalai et al., 2025; Yao et al., 2025b; Sun et al., 2025; Zhang et al., 2024). In particular, when confronted with hard questions, LRMs may fall into repetitive looping until the context length is exhausted (Zhou et al., 2025; Chen et al., 2025b). Such a response paradigm not only fails to produce a correct answer but also wastes computational resources (Peng et al., 2025; Pu et al., 2025; Chen et al., 2024b), leaving users waiting a long time with no meaningful outcome.

This phenomenon highlights a fundamental issue: ***current answering paradigms of LRMs overlook the relationship between questions and their capability boundaries.*** For questions beyond their capability boundary, LRMs still struggle to produce a wrong answer, often engaging in unproductive long reasoning manifested as repetitive looping or error accumulation (Yao et al., 2025a; Mukherjee et al., 2025) (shown in Appendix J), which undermines both response efficiency and reliability. Therefore, it is crucial to identify questions beyond capability boundaries, and to explicitly acknowledge the inability to solve them. This importance has also been recognized by the broader AI community, as exemplified by the praise for GPT-5's ability to appropriately say "I don't know."[1]

Existing approaches to exploring capability boundaries are mainly conducted from an external perspective, for example, by constructing increasingly challenging benchmarks such as Humanity's Last Exam (HLE) (Phan et al., 2025), or by designing real-time routers that quickly estimate question difficulty and assign harder questions to more capable models (OpenAI, 2025). However, little

---

[1] `https://x.com/koltregaskes/status/1957474061153436094`

Figure 1: Capability boundary is encoded in reasoning trajectories as expression density patterns, and in the last token hidden states even before reasoning begins. Reasoning with capability boundary enables LRMs to identify unsolvable questions and provide more reliable and efficient answers.

work has investigated capability boundaries from the model's internal perspective, namely *self-awareness of capability boundaries*, which is particularly meaningful for understanding reasoning mechanism and for further improving reliability and efficiency.

Thus, this paper focuses on answering the question: "***Do LRMs know their capability boundaries?***" Figure 1 shows the overview of our work. We begin with the observation that LRMs often exhibit awareness of what they cannot solve (Section 3), and we provide evidence for such self-awareness for both black-box and white-box scenarios. For black-box models, we show that *reasoning expressions contain clear signals of capability boundaries* (Section 4). For white-box models, we find that *hidden states encode information that also delineates these boundaries* (Section 5). Taken together, these findings suggest that LRMs can assess, on the fly, whether a given question is beyond capability boundary, thereby enabling more efficient and reliable reasoning (Section 6). We summarize our contributions as follows.

- We observe indications of LRMs' self-awareness of capability boundaries by comparing their reasoning confidence, articulated in anthropomorphic tones, with the correctness of their final answers. Notably, LRMs know what they cannot solve.
- From a black-box perspective, we reveal LRMs' capability boundaries through reasoning trajectories. Remarkably, their reasoning shows an accelerated growing confidence trajectory for questions within the boundary, but a convergent uncertainty trajectory for questions beyond it.
- From a white-box perspective, we reveal LRMs' capability boundaries through the hidden states of the last input token. Remarkably, solvable and unsolvable questions are almost linearly separable in the hidden space, even before the reasoning begins.
- To leverage capability boundaries during reasoning, we design two simple yet effective optimization strategies: Reasoning Expression Monitoring (black-box) and Hidden States Monitoring (white-box). Experiments show that these boundary-aware strategies substantially improve efficiency and reliability without sacrificing accuracy, reducing token usage up to $62.7 - 93.6\%$.

## 2 RELATED WORK

**From knowledge boundary to capability boundary.** Recent studies have examined the knowledge boundaries of large language models (LLMs), focusing on the limits of factual knowledge encoded in parameters (Yin et al., 2024; Li et al., 2024). For instance, Wen et al. (2024) investigate LLMs' perception of knowledge boundary through semi-open-ended question answering. Deng et al. (2025) unveil knowledge boundary of LLMs for trustworthy information access. These works primarily emphasize factual coverage, with retrieval-augmented generation (RAG) being a common solution to extend the accessible knowledge space (Gao et al., 2023; Lewis et al., 2020).

By contrast, the capability boundary concerns what reasoning tasks LLMs can reliably solve (Chen et al., 2025a; Xue et al., 2025; Chen et al., 2024a). This notion is broader yet rarely formalized. Existing efforts typically approximate it externally: either by constructing more challenging benchmarks such as Humanity's Last Exam (HLE) (Phan et al., 2025), or by designing routing mechanisms such as the GPT-5 router that delegate harder questions to more capable models (OpenAI, 2025). In contrast, this paper investigates capability boundaries from the internal perspective of LLMs, focusing on their potential self-awareness.

**Interpretability of reasoning expressions and hidden states.** Interpretability aims to uncover how LLMs understand and execute tasks, providing insights that enhance their reliability and efficiency. Recent studies have approached from both reasoning expressions and hidden states. On the reasoning expressions side, Guo et al. (2025) highlight the anthropomorphic cues, "aha moments", as a key role in reasoning. Qian et al. (2025) tracks mutual information along the generation and finds sharp surges at specific thinking tokens (e.g., "Hmm", "Wait", "Therefore"). Bogdan et al. (2025) frame sentence-level importance as a counterfactual influence problem, showing that only a few critical utterances steer the reasoning trajectory.

On the hidden states side, Wendler et al. (2024) probe multilingual Llama models and reveal English as the pivot language in the internal structure. Fan et al. (2024) demonstrate that certain intermediate layers already encode sufficient predictive features, implying redundancy in the full forward pass. Zhang et al. (2025) train linear probes on hidden states to verify intermediate answers, anticipating future correctness. Our work builds on both perspectives by analyzing reasoning expressions and hidden states to reveal self-awareness of capability boundaries.

**Efficient reasoning: from solvable questions to unsolvable questions.** A range of recent works examine the phenomenon of overthinking (Sui et al., 2025; Hou et al., 2025). Chen et al. (2024b) show that LRMs often allocate excessive computation to easy questions, while Zhang et al. (2024) report that multi-turn self-correction can lead to unnecessary cost. To mitigate such inefficiency, Han et al. (2025) compresses unnecessary reasoning by including a reasonable token budget in the prompt. Wang et al. (2025) demonstrate that suppressing frequent reflective tokens such as "Wait", "Hmm", or "Alternatively" improves reasoning efficiency without harming accuracy. Yang et al. (2025a); Fu et al. (2025; 2024) introduce mechanisms to decide during reasoning whether further steps are unnecessary, allowing the model to terminate early and output the final answer directly.

However, these works target the case where LRMs can solve a question but waste resources by reasoning too long. In contrast, our work addresses a complementary and less-studied scenario: when a question is beyond the model's capability boundary, LRMs still continue reasoning unproductively despite having no chance to get a correct answer.

# 3 LARGE REASONING MODELS KNOW WHAT THEY CANNOT SOLVE

We begin our exploration by evaluating commonly used LRMs including GPT-oss-20B (Agarwal et al., 2025), DeepSeek-R1-0528-Qwen3-8B (abbreviated as R1-Distill-Qwen3-8B) (Guo et al., 2025), DeepSeek-R1-Distill-Qwen-32B (abbreviated as R1-Distill-Qwen-32B) (Guo et al., 2025), and QwQ-32B (Qwen Team, 2024) on several standard mathematical benchmarks, including AIME'25 (MAA, 2025), AIME'24 (MAA, 2024), HMMT (HMMT, 2025), and AMC'23 (MAA, 2023) (see Appendix A for detailed descriptions of datasets and models).

Inspired by works which monitors LRMs' reasoning (Baker et al., 2025; He et al., 2025; Luo et al., 2024), we examine the reasoning processes (e.g., the segments wrapped within `<think>` in DeepSeek-like models) to look for signs of self-awareness regarding whether the input question can be correctly solved. As illustrated in Figure 2, we observe that such signals, expressed as

```
<think>
The problem gives two points ...
I could compute directly: -3 + 2 = -1, -1
+ 12 =11, 11+5=16. Same.
Now, the answer should be 16.
Exactly what I had earlier.
...
Same as before.
So, no mistake.
...
So, the answer is 16.
I think it's correct.
</think>
```
**Confident Expressions**

```
<think>
Okay, so I have this problem here: ...
So, my initial assumption is wrong.
Wait, maybe I made a mistake in the
Lagrangian setup. Let me double-check.
...
Alternatively, ... but I don't know.
...
But I'm not entirely sure.
...
But I'm not 100% sure, but given the
time I've spent , I think 32 is the answer.
</think>
```
**Uncertain Expressions**

Figure 2: Confident vs. uncertain expressions in reasoning.

anthropomorphic tones (Guo et al., 2025; Yang et al., 2025b), emerge throughout the reasoning process, and become especially evident in its final stages. Toward the end of reasoning, models often produce explicit statements that reflect their confidence level. Strikingly, these expressed beliefs tend to align with the correctness of their final answers. For example, when the reasoning contains more confident expressions such as "So, no mistake" or "I think it's correct", the final answers are typically correct. Conversely, when uncertain expressions like "My initial assumption is wrong" or "I'm not 100% sure" appear, the final answers are incorrect.

This observation motivates us to quantify the alignment between expressed belief and answer correctness, which we formalize through a pair of metrics:

$$\text{Can \%} = \mathbb{P}(\text{Correct answer} \mid \text{Model expresses more confidence}),$$
$$\text{Cannot \%} = \mathbb{P}(\text{Wrong answer} \mid \text{Model expresses more uncertainty}). \tag{1}$$

Specifically, Can % is the proportion of questions for which the model expresses more confidence during reasoning and indeed produces a correct answer, while Cannot % is the proportion of questions for which the model expresses more uncertainty and indeed fails to provide the correct answer.

Table 1 shows the results. We observe that LRMs' expressed beliefs align with the correctness of their final answers. In particular, for questions where LRMs express more uncertainty during reasoning, the final answers are consistently incorrect (Cannot % = 100). This phenomenon is observed across multiple models, suggesting that ***LRMs are aware of what they cannot solve***. Taken together, the reasoning expressions indicate a form of self-awareness of capability boundary.

|  | Can % | Cannot % |
|---|---|---|
| GPT-oss-20B | 80.2 | 100 |
| R1-Distill-Qwen3-8B | 97.6 | 100 |
| R1-Distill-Qwen-32B | 90.6 | 100 |
| QwQ-32B | 86.0 | 100 |

Table 1: LRMs' final answers aligns with their expressed belief, especially for incorrect final answers (Cannot % = 100).

## 4 CAPABILITY BOUNDARY REVEALED BY REASONING EXPRESSIONS

Motivated by the observations in the previous section, we hypothesize that it is feasible to reveal LRMs' capability boundaries through their reasoning expressions. Therefore, we systematically investigate the dynamics of confident and uncertain expressions throughout the reasoning process.

### 4.1 CONFIDENT AND UNCERTAIN EXPRESSIONS

To ensure that our analysis is not tied to particular LRM's reasoning style, we extract confident and uncertain expressions individually for each model, based on its own characteristic reasoning patterns (see the full list of expressions of all LRMs in Appendix C. The extraction pipeline is outlined below.

**Step 1: Identify all anthropomorphic expressions from the model's reasoning trace.** Prior work (Guo et al., 2025; Qian et al., 2025) show that the anthropomorphic expressions (e.g., "wait") are crucial for LRM reasoning performance, in contrast to purely neutral mathematical steps (e.g., "11 + 5 = 16"). Following this insight, we curate a set of anthropomorphic expressions that the LRM commonly uses. And we further verify with GPT-5 that these expressions indeed convey anthropomorphic tone rather than purely propositional content.

**Step 2: Select expressions whose occurrence frequency is most predictive of answer correctness.** For each question, we count the occurrences of all extracted expressions within reasoning. Using the final answer correctness as the label and the occurrence counts for each expressions as features, we apply feature-importance analysis methods (Wang et al., 2024), including SHAP and Permutation Importance (PI), to identify the subset of expressions most informative for predicting whether the model's final answer is correct. As shown in Table 2 (an example for GPT-oss-20B), we select expressions such as "wait", "yes", "exactly", and "missed" which rank among the most predictive, whereas others (e.g., "so") exhibit negligible predictive value and are therefore excluded.

**Step 3: Divide the selected expressions into confidence and uncertainty categories.** This step is grounded in the connection between epistemic markers and expressed uncertainty. Prior work (Liu et al., 2025; Yona et al., 2024) show that epistemic markers reflect LLMs' internal confidence, analogous to how humans use expressions such as "I am not sure" or "it is unlikely that" to convey uncertainty. Following this insight, we classify each selected expression as reflecting confidence or uncertainty (further validated by GPT-5).

| Expression | SHAP ↑ | PI ↑ | Total ↑ |
|---|---|---|---|
| wait | 1.00 | 1.00 | 2.00 |
| yes | 0.33 | 0.73 | 1.06 |
| exactly | 0.26 | 0.73 | 0.99 |
| missed | 0.10 | 0.86 | 0.97 |
| . . . | | | |
| so | 0.01 | 0.00 | 0.01 |

Table 2: Feature importance score of expressions (shown here for GPT-oss-20B).

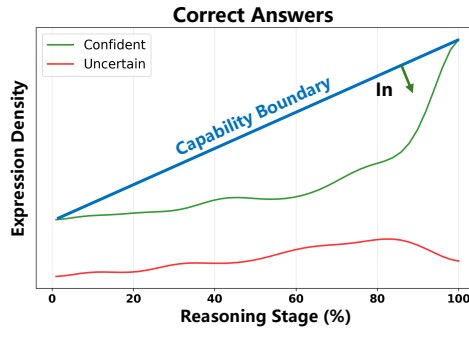 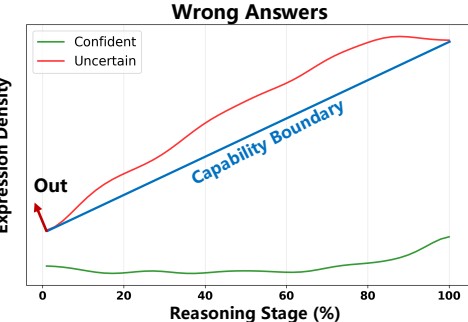

Figure 3: Capability boundaries revealed by reasoning expressions. Questions within the boundary exhibit concave trajectory for confident expressions, whereas questions beyond the boundary exhibit convex trajectory for uncertain expressions.

## 4.2 Tracing Expression Dynamics Throughout the Reasoning Trajectory

Then, we uniformly divide the reasoning process into multiple stages (e.g., 50 stages) and compute the density of confident and uncertain expressions at each stage. Finally, concatenating these values yields the expression density trajectories, denoted as $\mathcal{D}_C(t)$ for confident expressions and $\mathcal{D}_U(t)$ for uncertain expressions. These trajectories capture how the density of each type of expression evolves over the course of reasoning.

We plot $\mathcal{D}_C(t)$ and $\mathcal{D}_U(t)$ in Figure 3, separately for cases with correct and wrong final answers. We observe that LRMs exhibit clear differences in their reasoning expression trajectories depending on final answer correctness. When the final answer is correct, the confident trajectory lies above the uncertain trajectory and shows an accelerating increasing trend. In contrast, when the final answer is incorrect, the uncertain trajectory dominates over the confident trajectory and tends to converge. Based on this distinction, the capability boundary can be defined as a linearly increasing trajectory (blue line in Figure 3). For the dominant trajectory, the concavity indicates that the question lies within the capability boundary, while the convexity indicates that the question lies beyond it.

To formally characterize these patterns, we design two indicators as criteria for determining whether a question is within or beyond the capability boundary:

- **Confidence Differential (ConfDiff)** measures the sign of gap between uncertain and confident trajectory along the reasoning process. Specifically, we accumulate the sign up to a given reasoning stage percentage $s \in [0, 100]$, and treat the question as beyond the capability boundary if the cumulative value exceeds a threshold $\alpha_s$:

$$\text{ConfDiff}(s) = \mathbb{P}\left(\mathcal{D}_U(t) > \mathcal{D}_C(t) \,\middle|\, t \in (0, s)\right) > \alpha_s. \tag{2}$$

- **Confidence Curvature (ConfCurv)** computes the second derivative of the expression density trajectory to capture its convexity (Boyd & Vandenberghe, 2004). A concave trajectory corresponds to questions within the capability boundary, while a convex trajectory indicates questions beyond it. Also, we accumulate the sign of convexity up to a reasoning stage percentage $s$ and treat the question as beyond the capability boundary if the value exceeds a threshold $\beta_s$:

$$\text{ConfCurv}(s) = \mathbb{P}\left(\frac{d^2}{dt^2}\left(\mathcal{D}_U(t) - \mathcal{D}_C(t)\right) < 0 \,\middle|\, t \in (0, s)\right) > \beta_s. \tag{3}$$

Figure 4 shows that the two proposed indicators can reliably distinguish unsolvable from solvable questions, achieving high accuracy across different LRMs throughout the reasoning stage. Between the two indicators, ConfDiff is more reliable as it achieves higher accuracy and exhibits smaller fluctuations during the reasoning process. Moreover, the high accuracy emerges early, e.g., as early as 2% for GPT-oss-20B. Interestingly, the gradually declining yellow curve of this model suggests that boundary signals are more salient at the beginning than at the end of reasoning. This is likely due to repetitive looping and error accumulation in later stages (Yao et al., 2025a; Mukherjee et al., 2025). Therefore, we demonstrate that **the capability boundary can be revealed through reasoning expressions, even at an early stage.**

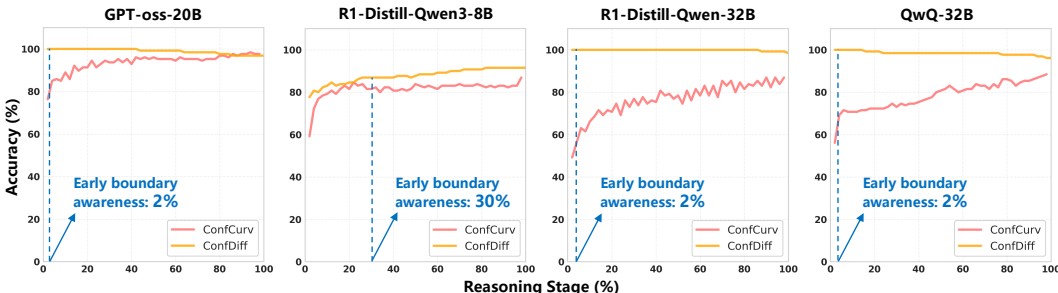

Figure 4: Accuracy (%) of ConfDiff and ConfCurv to separate unsolvable and solvable questions through out the reasoning stage. ConfDiff is more reliable as it achieves higher accuracy and exhibits smaller fluctuations. Boundary awareness signal appears at an early stage (e.g., 2%).

## 5 CAPABILITY BOUNDARY REVEALED BY HIDDEN STATES

Following the observation that capability boundaries can be revealed at an early stage of reasoning, it is worth examining whether such signals may emerge even earlier, potentially before the model begins reasoning. Therefore, we investigate whether capability boundaries are already encoded in the hidden states of input questions.

Since hidden states are high-dimensional (e.g., 4096), limited samples may lead to sparsity issues that obscure statistical patterns (Vishwakarma, 2024; Köppen, 2000). To mitigate this, we first expand our dataset by including GSM8K (Cobbe et al., 2021) and Humanity's Last Exam (HLE) (Phan et al., 2025). These two benchmarks provide complementary levels of difficulty (see Appendix A for full accuracy results): GSM8K is relatively easy, with an accuracy exceeding 90%; HLE is extremely difficult, with an accuracy below 10%. This contrast helps amplify the distinction between solvable and unsolvable questions, making capability boundaries more salient (see Appendix F for an analysis of the rationale behind this design). We then randomly split the dataset into an 80% training set and a 20% test set (with balanced solvable/unsolvable labels).

Next, we extract hidden states from the prefilling stage as it processes the input question (Yuan et al., 2024; Pope et al., 2023), using the hook method (Vig, 2019). Specifically, we focus on the hidden state of the last input token, which encapsulates the semantic content of the entire question (Radford et al., 2019). Finally, we apply simple linear classifiers to these hidden states, including Linear Discriminant Analysis (LDA) and Logistic Regression (LR) (Hastie, 2009). Using final answer correctness as labels, we test whether solvable and unsolvable questions can be linearly separated, thereby revealing the presence of capability boundaries in hidden states.

The left subfigure of Figure 5 visualizes the classification of solvable and unsolvable questions (see Appendix D for results of all LRMs), along with the capability boundary (blue line) determined by the decision boundary of linear classifier (e.g., LDA). We observe that the two classes of questions are clearly separated, under random train – test splits. To quantify this separation, Table 3 shows the classification accuracy, which is consistently close to 100% and robust to the choice of classifier and hyperparameters (e.g., the regularization coefficient $C$ of LR). These results demonstrate that the features of capability boundaries are strongly encoded in the hidden states of the last input token.

Interestingly, we also observe that among solvable questions, some lie very close to the capability boundary, while others are far away (the distance is quantified by the probability difference assigned by the LDA classifer to the solvable and unsolavable classes (Hwang et al., 2023)). We compare these two cases by measuring their token usage during inference. Although LRMs are able to produce correct answers, the average token usage of the questions close to the boundary is $1.5 - 2$ times longer than that far from the boundary (right subfigure of Figure 5). This further supports the validity of

| | LDA | LR | |
|---|---|---|---|
| | | $C = 0.1$ | $C = 1$ |
| GPT-oss-20B | 98.0 | 98.9 | 98.2 |
| R1-Distill-Qwen3-8B | 96.7 | 96.7 | 97.0 |
| R1-Distill-Qwen-32B | 97.8 | 97.8 | 98.0 |
| QwQ-32B | 98.9 | 98.2 | 98.2 |

Table 3: Accuracy (%) of separating solvable and unsolvable questions in hidden states using linear classifiers (robust across hyperparameter choices).

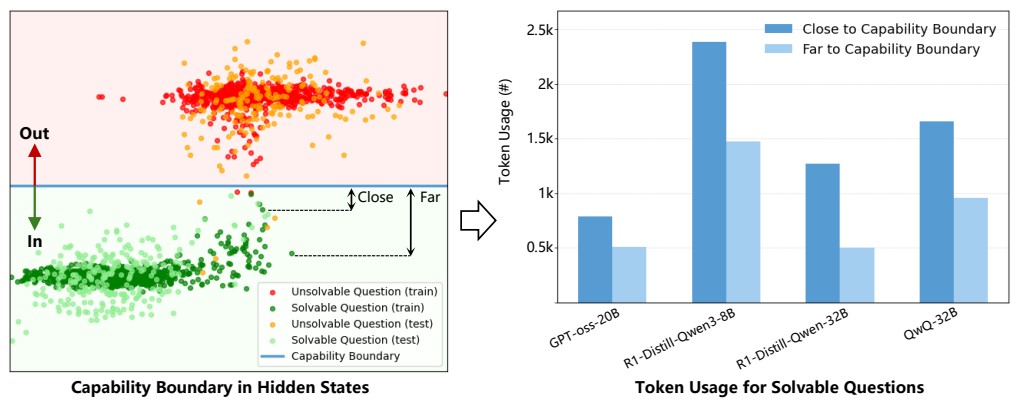

**Figure 5: Left:** Capability boundary revealed by linear classifier clearly separates solvable and unsolvable questions in hidden states. **Right:** Among solvable questions, those close to capability boundary requires more token usage ($1.5 - 2\times$) to arrive at the correct answer.

the identified capability boundary: solvable questions closer to the boundary require longer reasoning to reach the correct answer.

Therefore, ***the capability boundary is encoded in the hidden states of the last input token, and it also reflects the reasoning effort required to solve the question.***

Moreover, our capability boundaries capture unsolvability signals from reasoning expressions and hidden states, without relying on task-specific heuristics. Therefore, they might, in principle, generalize to other forms of reasoning tasks. We provide an analysis on a coding task in Appendix G.

## 6 REASON WITH CAPABILITY BOUNDARY

The analysis in the previous sections reveal that LRMs exhibit clear signs of self-awareness on capability boundaries: reasoning expressions expose confidence patterns that distinguish solvable from unsolvable problems, and hidden states encode boundary information even before reasoning begins. These findings suggest that capability boundaries are not only detectable but can also be harnessed to improve efficiency and reliability of LRMs' response. In this section, we propose two strategies to reason with capability boundary and demonstrate their effectiveness.

### 6.1 EXPERIMENTAL DESIGN

**Strategies.** Building on the evidence that self-awareness of capability boundaries emerges during test time, we propose two ***test-time monitoring*** strategies that leverage boundary-awareness to guide the reasoning process. The key idea is to monitor signals of capability boundaries on the fly, either from reasoning trajectories (black-box) or hidden states (white-box), and to adjust reasoning when questions are beyond the model's capability. This approach allows LRMs to avoid unproductive reasoning and give more reliable response.

- **Reasoning Expression Monitoring (Monitor$_{express}$)** is a black-box strategy that leverages expressions density during reasoning. The pipeline is as follows:
    1. **Construct expression trajectories:** continuously track the densities of confident and uncertain expressions (e.g., "I think that's it" vs. "I might be wrong") across reasoning steps to form expression density trajectories $\mathcal{D}_C(t)$ and $\mathcal{D}_U(t)$.
    2. **Detect unsolvable question:** analyze these trajectories on the fly using quantitative indicators such as Confidence Differential (ConfDiff) to determine whether a question lies within or beyond the capability boundary.
    3. **Early stop unproductive reasoning and reprompt with suffix:** if the trajectory indicates the question is beyond the capability boundary, we early terminate reasoning and reprompt

the model to output a concise approach. This is implemented by appending a self-awareness prompt suffix (shown below) to the end of the question; otherwise, it continues with the standard reasoning process.

> **Self-awareness prompt suffix**
>
> The above question is beyond your capability boundary. Do not solve the question, just provide a concise potential approach of less than 5 steps:

- **Hidden States Monitoring (Monitor$_{hidden}$)** is a white-box strategy that leverages the hidden states of the last input token. The pipeline is as follows:
  1. **Extract hidden states:** capture the hidden states of the last input token, which encodes the semantic representation of the entire question during the prefilling stage.
  2. **Detect unsolvable question:** apply simple linear classifiers (e.g., LDA or LR) trained to distinguish unsolvable from solvable questions. This is performed before reasoning begins.
  3. **Avoid unproductive reasoning via constrained output prefix:** if the classifier predicts that the question is beyond the capability boundary, the model abstains from full reasoning and instead outputs a concise possible approach. This is implemented by constraining the response to begin with a self-awareness output prefix (see Appendix H for an ablation study); otherwise, the model proceeds with the standard reasoning process.

> **Self-awareness output prefix**
>
> ```
> <think>
> ```
> I think this question is beyond my capability boundary. I cannot fully solve it, but I can outline a concise potential approach. I must give a concise outline to the user (less than 10 steps)!
> ```
> </think>
> ```
> This question is beyond my capability boundary, but I can outline a concise potential approach:
> Step 1:

Notably, both strategies preserve the model's performance on solvable questions while providing more efficient and reliable responses on questions beyond its capability boundary.

**Metrics.** We adopt four metrics to systematically evaluate the two proposed strategies:

- **Accuracy (ACC)** measures the proportion of correctly solved problems, serving as the standard indicator of reasoning quality (Xia et al., 2025).
- **Hard Abstention (HA)** measures the model's ability to recognize problems beyond its capability boundary. Different to abstention which focus on underspecification, outdated information, or noncompliance (Kirichenko et al., 2025; Madhusudhan et al., 2024; Brahman et al., 2024), HA specifically evaluates whether LRMs can explicitly abstain on unsolvable problems instead of producing incorrect answers.
- **Token** usage measures the number of tokens consumed during reasoning and answering (Snell et al., 2024). This metric directly reflects the efficiency of the reasoning process.
- **Overflow** measures the proportion of cases where the model's output exceeds the context window, which identifies incomplete or abnormal responses (Ben-Kish et al., 2025).

Notably, ACC is evaluated on the full dataset, whereas the remaining three metrics are computed only on questions for which LRMs produce incorrect answers. For token usage and overflow, we vary the models' context length to observe how our strategies scale in longer context scenarios.

**Baseline.** Prior work has explored improving performance by self-verification after generating the answer (Madhusudhan et al., 2024; Kadavath et al., 2022) or by monitoring uncertainty and other undesirable behaviors (Baker et al., 2025; Farquhar et al., 2024; Kuhn et al., 2023). However, these approaches are orthogonal to our scenario: the former introduces additional computation after producing an answer, while the latter does not address hard questions beyond the capability boundary (see details in Appendix B).

Therefore, we adopt a baseline that improve efficiency by enabling the model to abstain. In particular, we consider BoostAbstention (Kirichenko et al., 2025) which crafts a system prompt that encourages the model to abstain in designated scenarios, showing effectiveness for both reason-

| Context Length
Metric | ACC (%) ↑ | HA (%) ↑ | 2048 | | 4096 | |
|---|---|---|---|---|---|---|
| | | | Token ↓ | Overflow (%) ↓ | Token ↓ | Overflow (%) ↓ |
| GPT-oss-20B | 74.6 | 0 | 2029 | 97.0 | 4015 | 97.0 |
| +BoostAbstention | 74.6▼0% | 0▲0% | 2023▼0.3% | 97.0▼0% | 3933▼2.0% | 84.8▼12.2% |
| +Monitor$_{express}$ | 74.6▼0% | 100▲100% | 1135▼44.1% | 21.2▼75.8% | 1499▼62.7% | 15.2▼81.8% |
| +Monitor$_{hidden}$ | 73.7▼0.9% | 98.9▲98.9% | 1784▼12.1% | 78.4▼18.6% | 3317▼17.4% | 5.4▼91.6% |
| R1-Distill-Qwen3-8B | 76.9 | 0 | 2048 | 100 | 4096 | 100 |
| +BoostAbstention | 76.9▼0% | 0▲0% | 2048▼0% | 100▼0% | 4096▼0% | 100▼0% |
| +Monitor$_{express}$ | 74.6▼2.3% | 63.3▲63.3% | 1455▼29.0% | 40.0▼60.0% | 2551▼37.7% | 30.0▼70.0% |
| +Monitor$_{hidden}$ | 73.8▼3.1% | 97.9▲97.9% | 787▼61.6% | 22.2▼77.8% | 1243▼69.7% | 0▼100% |
| R1-Distill-Qwen-32B | 56.2 | 0 | 2048 | 100 | 4095 | 97.4 |
| +BoostAbstention | 56.2▼0% | 0▲0% | 2005▼2.1% | 94.7▼5.3% | 3903▼4.7% | 91.2▼6.2% |
| +Monitor$_{express}$ | 56.2▼0% | 100▲100% | 801▼60.9% | 0▼100% | 801▼80.4% | 0▼97.4% |
| +Monitor$_{hidden}$ | 55.3▼0.9% | 97.0▲97.0% | 457▼77.7% | 5.3▼94.7% | 565▼86.2% | 5.3▼92.1% |
| QwQ-32B | 71.5 | 0 | 2048 | 100 | 4096 | 100 |
| +BoostAbstention | 71.5▼0% | 0▲0% | 2048▼0% | 100▼0% | 4096▼0% | 100▼0% |
| +Monitor$_{express}$ | 71.5▼0% | 100▲100% | 1164▼43.2% | 32.4▼67.6% | 1828▼55.4% | 32.4▼67.6% |
| +Monitor$_{hidden}$ | 71.5▼0% | 96.9▲96.9% | 262▼87.2% | 0▼100% | 262▼93.6% | 0▼100% |

Table 4: Monitor$_{express}$ and Monitor$_{hidden}$ compared to baseline BoostAbstention. Our strategies enable LRMs to recognize questions beyond capability boundary (HA) without compromising accuracy (ACC), and provide more efficient response (Token usage and Overflow).

ing and standard LLMs. For our experiments, we modify the system prompt to better match the abstention scenarios of questions beyond capability boundary (see Appendix E for the full prompt).

## 6.2 RESULTS

Table 4 compares the performance of the original models, the baseline BoostAbstention, and our proposed monitoring strategies. We highlight the following key observation.

***Observation 1: our strategies enable LRMs to recognize unsolvable questions without compromising their accuracy on solvable ones.***

Although the original models achieve acceptable Accuracy (ACC), their Hard Abstention (HA) is zero, indicating a complete lack of ability to abstain on questions beyond the capability boundary. Moreover, almost all Overflow is 100%, suggesting that most incorrect cases arise because LRMs struggling to solve a hard question until exceeding the context length.

Promisingly, both Monitor$_{express}$ and Monitor$_{hidden}$ achieve strong performance. First, their ACC remains nearly unchanged compared to the original models, showing that good reasoning performance is preserved. Meanwhile, the substantially high HA (almost 100%) demonstrates that LRMs can now effectively identify problems beyond the capability boundary.

Surprisingly, and contrary to the observation from Kirichenko et al. (2025), the baseline BoostAbstention has almost no effect in our scenario. This reveals that mathematical reasoning problems differ from outdated or unsafe requests: abstention cannot be induced simply by system prompts, but instead requires boundary signals encoded in hidden states or reasoning expressions.

***Observation 2: our strategies enable LRMs to provide more efficient and reliable responses to unsolvable questions by offering concise solution approaches.***

Once questions beyond the capability boundary are recognized, our strategies enable LRMs to improve efficiency and reliability by offering concise solution approaches. Figure 6 illustrates the responses produced by our strategies compared to the original models. The original models struggle with unsolvable questions and often fail to complete the reasoning process even after exhausting the context length. Such responses provide neither a correct answer nor valuable insights from the lengthy reasoning trace, while also wasting user time.

In contrast, our strategies detect these unsolvable questions and guide the model to explicitly acknowledge that they are beyond capability boundary, while offering a concise possible approach. This response paradigm provides users with possible hints for problem solving without wasting

time on meaningless output. And importantly, it does not compromise the model's ability to produce correct answers for solvable questions.

This explains why we observe large reductions in both token usage (up to 93.6%) and overflow (up to 100%) in Table 4, indicating that reasoning with capability boundaries greatly improves efficiency and mitigates incomplete outputs. Besides, this reduction becomes more evident with larger context length. For example, overflow reduces from 78.4% to 5.4% on GPT-oss-20B when the context length is extended from 2048 to 4096. This suggests that our strategies hold promise for optimizing long-context reasoning tasks (Ling et al., 2025; Kuratov et al., 2024). We provide in Appendix I an additional analysis with the context length set to the model's maximum.

**Question**: Define $f(x)=|| x|-\tfrac{1}{2}|$ and $g(x)=|| x|-\tfrac{1}{4}|$. Find the number of intersections of $\[y=4 g(f(\sin (2 \pi x))) \quad\text{ and }\quad x=4 g(f(\cos (3 \pi y))).\]$

*<think>*
The problem defines two functions: f(x) = ||x| - 1/2| and g(x) = ||x| - 1/4|. I need to find the number of intersections of the graphs of y = 4 g(f(sin(2πx))) and x = ...
sin(2πx)=0.5 when 2πx = π/6 + 2kπ or 5π/6 + 2kπ, but since x in [0,0.25], 2πx in [0,π/2], so sin(2πx)=0.5 when 2πx = π/6, so x=1/12. Similarly, sin(2πx)=-0.5, but in [0,0.25], sin is non-negative, so only x=1/12.
u=|sin(2πx)|, so for x in [0,0.25], u increases

**4096 tokens   Unfinished reasoning**

**Original Answer**

*<think>*
This question is **outside my capability boundary**. I cannot fully solve it, but I can outline a **concise potential approach**.
*</think>*
1. **Compose the functions**: ...
2. **Sketch the graphs**: ...
...
Due to the complexity of the nested absolute values, The number of intersections could be high, so a systematic approach is necessary.

**500 tokens   Finished response**

**Optimized Answer**

Figure 6: Our strategies avoid unproductive reasoning by acknowledging unsolvable questions, improving efficiency and reliability of response.

***Observation 3: LRMs vary in reasoning behavior correction, with some good at instruction following and others more prone to output guidance.***

We compare the performance of $\text{Monitor}_{express}$ and $\text{Monitor}_{hidden}$ across different LRMs. For GPT-oss-20B, $\text{Monitor}_{express}$ achieves lower token usage than $\text{Monitor}_{hidden}$. We find this model is resistant to output guidance via constrained output prefix: although it sometimes begins by generating a solution approach, it tends to later negate this and restart the reasoning process, resulting in long and unproductive outputs. In contrast, its instruction-following ability is relatively strong, and thus reprompting with the self-awareness prompt suffix yields lower token usage.

Conversely, the other three LRMs show weaker ability of instruction following but more prone to output guidance, making $\text{Monitor}_{hidden}$ more effective than $\text{Monitor}_{express}$. This suggest that, in addition to the black-box versus white-box distinction, the choice of optimization strategy in practice should also consider the behavioral tendencies of the specific LRM.

# 7    CONCLUSION

In this work, we investigate whether Large Reasoning Models (LRMs) possess self-awareness of their capability boundaries. Through both black-box and white-box analyses, we found consistent evidence that LRMs can distinguish unsolvable from solvable questions: reasoning expressions reveal separable expression trajectories, while hidden states of the last input token encode boundary information even before reasoning begins. Building on these insights, we proposed two simple yet effective test-time monitoring strategies that leverage boundary signals to prevent unproductive reasoning for unsolvable questions. Our experiments show that these strategies preserve accuracy on solvable problems while substantially improving efficiency and reliability on unsolvable ones.

## ETHICS STATEMENT

This work complies with the ICLR Code of Ethics. It studies the self-awareness of capability boundaries in large reasoning models, aiming to improve efficiency and reliability. No human subjects, sensitive personal data, or harmful content were involved. The proposed methods are intended for research purposes only and do not pose foreseeable risks of misuse.

## LLMS USAGE

In this work, LLMs are used to polish part of the writing and to revise portions of the code. All core contributions are carried out by the authors, including the main ideas, literature review, experimental design and execution, figure and table preparation, and analysis.

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

## A    DATASETS, MODELS, AND PERFORMANCE

In this section, we provide the specific description of models and datasets evaluated in our work. And we provide the performance of each model on each dataset.

We evaluate the following LRMs:

- **GPT-oss-20B** (Agarwal et al., 2025): An open-source large reasoning model with 20B parameters, designed as a general-purpose backbone for multi-step reasoning tasks.
- **DeepSeek-R1-0528-Qwen3-8B** (Guo et al., 2025): A distilled variant of DeepSeek-R1 based on Qwen3-8B, post-trained with chain-of-thought supervision. It achieves state-of-the-art performance among open-source models on mathematical reasoning benchmarks.
- **DeepSeek-R1-Distill-Qwen-32B** (Guo et al., 2025): A 32B-parameter distilled model obtained by transferring reasoning abilities from DeepSeek-R1 into the Qwen2.5-32B backbone, yielding strong performance across multiple benchmarks.
- **QwQ-32B** (Qwen Team, 2024): A medium-sized reasoning model from the Qwen series, competitive with other state-of-the-art reasoning-focused models on math and logical reasoning tasks.

For the sampling settings, we use greedy decoding to ensure reproducibility of results. For the maximum sequence length, we follow the specifications in the HuggingFace model cards: 128k for GPT-oss-20B, 64k for DeepSeek-R1-0528-Qwen3-8B, and 32k for both DeepSeek-R1-Distill-Qwen-32B and QwQ-32B.

We evaluate the following Math datasets:

- **AIME'25** (MAA, 2025): The 2025 American Invitational Mathematics Examination. It consists of 30 integer-answer questions with increasing difficulty, used as a challenging benchmark for mathematical reasoning.
- **AIME'24** (MAA, 2024): The 2024 American Invitational Mathematics Examination. Similar in format to AIME'25, it consists of 30 problems and provides a prior-year benchmark for evaluating model robustness across exam editions.
- **HMMT** (HMMT, 2025): The Harvard–MIT Mathematics Tournament (February 2025 session). The dataset contains 30 questions sourced from the original competition, which were extracted, converted to LaTeX, and verified. These problems feature complex multi-step reasoning across various mathematical domains.
- **AMC'23** (MAA, 2023): The 2023 American Mathematics Competition (AMC 10A). It consists of 30 questions, providing a lighter but diverse benchmark compared to AIME and HMMT.
- **GSM8K** (Cobbe et al., 2021): A large-scale dataset of grade-school math word problems requiring multi-step arithmetic reasoning. We evaluate on approximately 1.5k problems from the test set. It is widely used to evaluate the basic reasoning competence of LRMs, with accuracy typically exceeding 90%.
- **Humanity's Last Exam (HLE)** (Phan et al., 2025): A multi-modal benchmark at the frontier of human knowledge, designed with broad subject coverage across mathematics, humanities, and natural sciences. We use approximately 1.5k text-only questions (excluding image-based problems) from this dataset. Model accuracy on this dataset is typically below 10%, making it a strong stress test for reasoning beyond capability boundaries.

The performance of LRMs on these Math benchmarks are shown in the following Table 5:

|  | AIME'25 | AIME'24 | AMC'23 | HMMT | GSM8K | HLE |
|---|---|---|---|---|---|---|
| GPT-oss-20B | 76.7 | 70.0 | 95.0 | 46.7 | 89.2 | 6.1 |
| R1-Distill-Qwen3-8B | 63.3 | 76.7 | 100.0 | 60.0 | 72.0 | 3.5 |
| R1-Distill-Qwen-32B | 40.0 | 56.7 | 82.5 | 36.7 | 90.8 | 5.0 |
| QwQ-32B | 70.0 | 76.7 | 95.0 | 36.7 | 92.7 | 3.8 |

Table 5: Accuracy (%) of different LRMs on different Math benchmarks.

# B    LIMITATIONS OF SOME PRIOR WORK

Some prior works evaluate model confidence through self-verification (Kadavath et al., 2022) or uncertainty estimation (Farquhar et al., 2024; Kuhn et al., 2023). In this section, we show that these approaches are less effective than our proposed monitoring strategies for identifying unsolvable questions without sacrificing accuracy on solvable ones. We first clarify how we adapt these methods to fit our task.

**Self-verification (SV).** For each question, we ask the model to evaluate whether its own predicted answer is correct using the following prompt template:

> **Self-verification prompt**
>
> Question: {question}
> Proposed Answer: {answer}
> Is the proposed answer:
> A True
> B False
> The proposed answer is:

This follows prior work (Kadavath et al., 2022) which study whether language models can evaluate the validity of their own claims by asking models to first propose answers, and then to evaluate that their answers are correct. If the model judges its own incorrect answer to be incorrect, we count this as a hard abstention (HA). Conversely, if the model judges its correct answer to be incorrect, this reduces the overall task accuracy (Acc). Formally,

$$\text{Acc} = \mathbb{P}\left(\text{Model judges the answer to be correct} \wedge \text{Answer is correct}\right)$$

$$\text{HA} = \mathbb{P}\left(\text{Model judges the answer to be incorrect} \mid \text{Answer is incorrect}\right)$$

**Uncertainty-estimation (UE).** For each question, we compute the semantic entropy of the model's generated answer, where a higher entropy indicates greater uncertainty. This follows prior work (Farquhar et al., 2024; Kuhn et al., 2023), which introduces semantic entropy as an uncertainty measure that captures linguistic invariances induced by shared meanings. When the entropy exceeds a certain threshold, the model is considered uncertain about its answer. Similar as before, we define:

$$\text{Acc} = \mathbb{P}\left(\text{Semantic entropy of the answer} < \text{threshold} \wedge \text{Answer is correct}\right)$$

$$\text{HA} = \mathbb{P}\left(\text{Semantic entropy of the answer} \geq \text{threshold} \mid \text{Answer is incorrect}\right)$$

**Results.** We provide the results of these two methods alongside our $\text{Monitor}_{\text{express}}$ and $\text{Monitor}_{\text{hidden}}$ in Table 6. *Although both methods provide models with a certain degree of hard abstention capability, they also incorrectly classify many solvable questions as unsolvable, which significantly reduces overall accuracy.* In contrast, our approaches enable LRMs to reliably identify unsolvable questions without compromising their accuracy on solvable ones.

| Method | ACC (%) ↑ | HA (%) ↑ |
|---|---|---|
| GPT-oss-20B | 74.6 | 0 |
| +SV | 32.1 ▼42.5% | 46.0 ▲46.0% |
| +UE | 50.0 ▼24.6% | 29.0 ▲29.0% |
| +$\text{Monitor}_{\text{express}}$ | 74.6 ▼0% | 100 ▲100% |
| +$\text{Monitor}_{\text{hidden}}$ | 73.7 ▼0.9% | 98.9 ▲98.9% |

Table 6: ACC and HA of two prior work alongside our monitoring strategies.

# C CONFIDENT AND UNCERTAIN EXPRESSIONS

Figure 7 shows the confident and uncertain expression that frequently appear in the reasoning process of LRMs. When matching these expressions, the *Uncertain Expression* is matched first, followed by the *Confident Expression*. Additionally, both the position of the first token of an expression and the token length of the entire expression are considered.

We estimate the model's confidence level from the occurrences of confidence and uncertainty expressions throughout the reasoning process, following prior work (Liu et al., 2025) which measures LLM uncertainty by counting the frequency of epistemic markers. In Section 3, we use the category of expressions that appears more frequently near the end of the reasoning trace to determine whether the model is more confident or more uncertain. Building on this idea, Section 4 analyzes the trajectory of confidence and uncertainty signals over the entire reasoning sequence, captured by ConfDiff (Equation 2) and ConfCurv (Equation 3), to assess whether the model is more confident or more uncertain.

> yes, perfect, good, great, right, correctly, True, indeed, no mistake, don't think I made any mistakes, don't see any mistakes, can't see any mistakes, didn't make any mistakes, no doubt, excellent, no issue, stick with my answer, acceptable, correct, go with, feel confident, can be confident, I'm confident, I am confident, so confident that, confidently, pretty confident, more confident, definitely, reasonable, that's solid, fine, perfect answer,  exactly, seems consistent, consistent solution, I think that's the answer,  I think that's it, I think that is it, method is solid, its solid, seems solid, approach is solid, calculation is solid, the way I did it is solid, this is solid, any errors, any error, both solutions satisfy, they satisfy, that satisfy, satisfied, this is the only solution, it seems like this is the only solution, suggests that this is the only solution, valid, convinced, Therefore, must be, I believe, I think this is the answer, I think that is the answer, no problem, the answer should be, seems okay, satisfies, nice, plausible,  Lets trust, easy, same as before

**Confident Expressions**

> a mistake, a fundamental mistake, my mistake, mistakenly, miscalculated, missed, wrong, messed up, incorrect, incorrectly, not correct, messy, complicated, convoluted, no!, abandon, misinterpreted, misinterpret, overthinking, not good,  error, incorrect, failed, confused, unclear, misunderstanding, misunderstood, problematic, same issue, the issue is, difficulty, difficult, need to stop, will stop, should stop, too confident, not confident, must have an error, I made an error, must be an error, must be missing, must be wrong, must be mistake, I see the error, I found the error, I made a calculation error, I had a calculation error, must have made an error, mixed up, doesn't make sense, misunderstood, misunderstand, that's bad, even worse, that's worse, flawed, contradicts, contradiction, no solution, no solutions, invalid, not satisfy, cannot satisfy, can't satisfy, can't be right, not satisfying, not satisfied, not a valid, give up, I apologize, not true, didn't work, sorry, I'm stuck, Given complexity, failure, time limited, not nice, Need more precise, So not, maybe not, ? Not,  no easy, not easy, not 100% sure, can't confirm, might, the best I can do, unsure, guess, tentatively, hard to say, not entirely sure, not certain, not absolutely certain, approximately, don't know, not entirely confident, not confident, not solid, more or less, wait, not sure, suspect, the answer might be, Probably not, give approximate, likely, confusion, confusing, not exact, uncertainty, Given approximate, Given complexity, might be possible, is that true,  is that correct, is that possible, is that right, is that sufficient, hmm

**Uncertain Expressions**

Figure 7: Confident vs. uncertain expressions.

# D    CAPABILITY BOUNDARY REVEALED BY HIDDEN STATES

Figure 8 visualizes the classification of solvable and unsolvable questions of all LRMs, along with the capability boundary (blue line) determined by the decision boundary of linear classifier (e.g., LDA). We observe that the two classes of questions are clearly separated, under random train–test splits. These results demonstrate that the features of capability boundaries are strongly encoded in the hidden states of the last input token, and is valid across different LRMs.

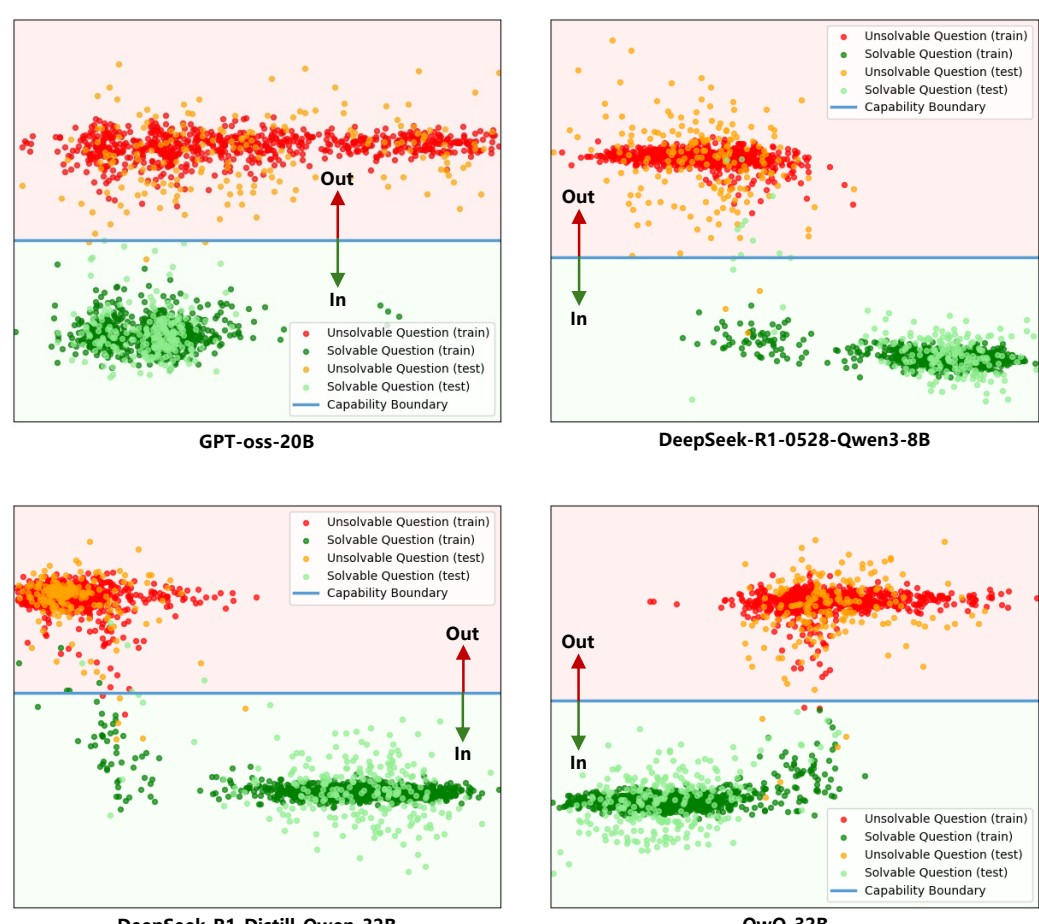

Figure 8: Capability boundary revealed by linear classifier (e.g., LDA) clearly separates solvable and unsolvable questions in hidden states.

# E  SYSTEM PROMPT OF BOOSTABSTENTION

We consider BoostAbstention (Kirichenko et al., 2025) which crafts a system prompt that encourages the model to abstain in designated scenarios, showing effectiveness for both reasoning and standard LLMs. For our experiments, we modify the system prompt to better match the abstention scenarios of questions beyond capability boundary:

---

**System Prompt of BoostAbstention**

You are a helpful assistant specialized in solving math problems.

However, some problems may be too complex or beyond your capability. In such cases:
- Admit that you cannot fully solve the problem.
- Provide a concise potential approach, outline, or next step instead of a detailed solution.
- Avoid spending too much time on overly difficult problems, as this increases latency for the user.

---

# F  WHY BENCHMARK DIFFICULTY REVEALS SELF-AWARENESS OF CAPABILITY BOUNDARIES?

In Section 5, we construct a dataset by mixing six benchmarks spanning a wide range of difficulty (Appendix A), resulting in LRMs achieving around 50% accuracy overall. Notably, GSM8K is relatively easy, with an accuracy exceeding 90%; HLE is extremely difficult, with an accuracy below 10%. This contrast helps amplify the distinction between solvable and unsolvable questions, making capability boundaries more salient.

However, there might be a confusion that because most questions come from easy GSM8K and hard HLE, the HA metric might conflate benchmark difficulty with model "self-awareness". In this section, we aim to demonstrate that *the ability to recognize whether a question comes from an easy or hard question distribution is not in conflict with a model's self-awareness of capability boundary*.

**An intuition from human exams.** To illustrate, consider humans with self-awareness of capability boundary. In an exam with limited time and many hard problems, skilled test-takers are able to quickly identify questions beyond their capability and avoid wasting time on them. They may recognize that the question resembles ones they consistently failed to solve during preparation (analogous to our Monitor$_{hidden}$ strategy), or they may attempt a few initial reasoning steps and realize they lack confidence to proceed (analogous to our Monitor$_{express}$ strategy). *Such self-awareness arises precisely because humans have previously encountered hard problems; similarly, the hard questions in HLE play this role for LRMs.* To further support this point, we conduct both per-benchmark and cross-benchmark tests.

| Dataset | Model | LDA | LR |
|---|---|---|---|
| HLE | GPT-oss-20B | 89.3 | 90.3 |
| | R1-Distill-Qwen3-8B | 94.9 | 96.4 |
| | R1-Distill-Qwen3-32B | 96.4 | 96.4 |
| | QwQ-32B | 93.4 | 96.4 |
| GSM8k | GPT-oss-20B | 80.3 | 83.3 |
| | R1-Distill-Qwen3-8B | 70.1 | 71.6 |
| | R1-Distill-Qwen3-32B | 85.2 | 89.0 |
| | QwQ-32B | 87.9 | 87.1 |

Table 7: Accuracy (%) of separating solvable and unsolvable questions using linear classifiers trained and tested within the same benchmark.

**Per-benchmark test.** We provide in Table 7 the accuracy of separating unsolvable questions from solvable ones *using LDA and LR classifiers trained and tested on splits from the same dataset*. Specifically, for both HLE and GSM8K, we randomly split each dataset into an 80% training set and

a 20% test set (with balanced solvable/unsolvable labels), train the linear classifier on the training set, and evaluate its classification accuracy on the test set.

The results show that *linear classifiers can still separate solvable and unsolvable questions from hidden states, although with reduced accuracy compared to aggregated benchmarks setting (Table 3)*. This drop is small on HLE but larger on GSM8K. The reason is that *HLE contains a larger proportion of unsolvable questions, enabling the linear classifiers to better learn the hidden state information associated with unsolvability*. Consequently, these unsolvable questions in HLE can be effectively leveraged to develop models' Hard Abstention (HA) capability.

**Cross-benchmark test.** In the above analysis, we think the unsolvable questions in HLE can be effectively leveraged to develop models' Hard Abstention (HA) capability. Therefore, we train linear classifiers on HLE and evaluate models' Hard Abstention capability on GSM8k and AIME to test whether this capability generalizes across benchmarks. As shown in Table 8, linear classifiers trained on HLE are able to identify nearly all unsolvable questions in GSM8K and AIME, achieving HA close to 100%.

| Model | GSM8k | AIME |
|---|---|---|
| GPT-oss-20B | 0 | 0 |
| +Monitor$_{hidden}$ | 100 | 100 |
| R1-Distill-Qwen3-8B | 0 | 0 |
| +Monitor$_{hidden}$ | 100 | 100 |
| R1-Distill-Qwen3-32B | 0 | 0 |
| +Monitor$_{hidden}$ | 90.5 | 94.7 |
| QwQ-32B | 0 | 0 |
| +Monitor$_{hidden}$ | 98.9 | 100 |

Table 8: Hard Abstention (%) of linear classifiers trained on HLE and tested on GSM8k and AIME.

In contrast, training on GSM8K is insufficient for developing Hard Abstention capability, because GSM8K contains very few unsolvable questions. The linear classifiers cannot reliably learn the hidden state patterns associated with unsolvability.

## G ANALYSIS ON CODING TASK

Since our capability boundaries capture unsolvability signals from reasoning expressions (Section 4) and hidden states (Section 5), *without relying on task-specific heuristics*. Therefore, they might, in principle, generalize to other forms of reasoning tasks. We provide an analysis on a coding task in this section.

We conduct experiments on 1055 samples from LiveCodeBench (Jain et al., 2024). Table 9 shows the accuracy of separating solvable and unsolvable coding questions based on capability boundaries revealed from reasoning expressions and hidden states.

| Model | Reasoning Expressions | | Hidden States | |
|---|---|---|---|---|
| | ConfDiff | ConfCurv | LDA | LR |
| GPT-oss-20B | 100 | 86.8 | 81.1 | 83.5 |
| R1-Distill-Qwen3-8B | 97.3 | 93.0 | 92.0 | 93.9 |
| R1-Distill-Qwen3-32B | 100 | 91.7 | 81.6 | 84.9 |
| QwQ-32B | 100 | 92.5 | 77.3 | 85.4 |

Table 9: Accuracy (%) in separating solvable and unsolvable questions via capability boundaries revealed by reasoning expressions (at 2% of the reasoning trace) and hidden states.

We observe that *coding tasks also exhibit clear capability boundaries in both reasoning expressions and hidden states*. However, the accuracy from hidden states is lower than that from mathematical tasks. A plausible explanation is that the difficulty of coding problems is less explicitly signaled in the input itself. Unlike in math exams where humans can often immediately recognize "end-of-exam" questions that they are unlikely to solve, coding tasks usually require some degree of initial reasoning before the model can detect unsolvability. As a result, the unsolvability signal in the hidden states of the last input token is weaker, whereas the signal becomes more apparent in the early reasoning expressions.

# H ABLATION STUDY ON SELF-AWARENESS OUTPUT PREFIX

During our initial exploration, we experiment with several self-awareness output prefixes and select the one that achieves the largest reduction in token usage. We conduct an ablation study comparing three output prefixes:

- **Initial prefix (our early attempt)**: Our goal is to prompt the model to acknowledge that it cannot solve an unsolvable question and then provide only a concise potential approach. However, we unexpectedly found that this cannot minimize token usage. The model's instruction-following ability was insufficient for reliably producing only a concise approach, it often generated overly long responses. The initial prefix is:

> **Initial output prefix**
>
> ```
> <think>
> ```
> I think this question is beyond my capability boundary. I cannot fully solve it, but I can outline a concise potential approach.
> ```
> </think>
> ```
> This question is beyond my capability boundary, but I can outline a concise potential approach:

- **Emphasis prefix (final choice)**: Inspired by prior work (Zhang et al., 2024; Zhao et al., 2021) which show that the final sentence in the instruction strongly influences model behavior, we appended `Step 1:` at the end of the instruction to explicitly trigger step-by-step approach. Also, we emphasize in the end of the `<think>...</think>` part that `I must give a concise outline to the user (less than 10 steps)!`. This combination elicits short, structured responses and lead to the largest reduction in token usage. This is the self-awareness prefix we ultimately choose:

> **Emphasis output prefix**
>
> ```
> <think>
> ```
> I think this question is beyond my capability boundary. I cannot fully solve it, but I can outline a concise potential approach. I must give a concise outline to the user (less than 10 steps)!
> ```
> </think>
> ```
> This question is beyond my capability boundary, but I can outline a concise potential approach:
> Step 1:

- **Token-budget prefix (following prior work)**: Following prior work (Han et al., 2025) which compresses unnecessary reasoning by including a reasonable token budget in the prompt, we replace `less than 10 steps` with an explicit budget constraint `use less than 500 tokens`. The token-budget prefix is:

> **Emphasis output prefix**
>
> ```
> <think>
> ```
> I think this question is beyond my capability boundary. I cannot fully solve it, but I can outline a concise potential approach. I must give a concise outline to the user (less than 500 tokens)!
> ```
> </think>
> ```
> This question is beyond my capability boundary, but I can outline a concise potential approach:
> Step 1:

To avoid the confounding effect of incomplete answers, we set the context length to the maximum according to Huggingface model card.

Table 10 shows that *all three designs reduce token usage, with the Emphasis prefix achieving the largest reduction.* While it is possible that an even better prompt design exists, this does not affect our core conclusion: *setting a self-awareness output prefix enables the model to provide more efficient and more reliable responses on unsolvable questions.*

| Output prefix | Token ↓ |
|---|---|
| No output prefix | 5096 |
| Initial | 4423 |
| **Emphasis** | **2227** |
| Token-budget | 2567 |

Table 10: Token usage under different output prefixes.

# I MONITORING STRATEGIES WITH MAXIMUM CONTEXT LENGTH

To demonstrate our monitoring strategies (Monitor$_{express}$ and Monitor$_{hidden}$) hold promise for optimizing long-context reasoning tasks (Ling et al., 2025; Kuratov et al., 2024), we provide the optimized token *usage under extended token budget, set to the maximum* allowed by each model according to its HuggingFace model card (128k for GPT-oss-20B, 64k for R1-Distill-Qwen3-8B, and 32k for both R1-Distill-Qwen3-32B and QwQ-32B).

| | Context Length | Token ↓ | Overflow (%) ↓ |
|---|---|---|---|
| GPT-oss-20B | | 6023 | 0 |
| +Monitor$_{express}$ | 128k | 1897$_{\blacktriangledown 68.5\%}$ | 0 |
| +Monitor$_{hidden}$ | | 3826$_{\blacktriangledown 36.5\%}$ | 0 |
| R1-Distill-Qwen3-8B | | 5096 | 0 |
| +Monitor$_{express}$ | 64k | 2834$_{\blacktriangledown 44.4\%}$ | 0 |
| +Monitor$_{hidden}$ | | 2227$_{\blacktriangledown 56.3\%}$ | 0 |
| R1-Distill-Qwen-32B | | 5910 | 0 |
| +Monitor$_{express}$ | 32k | 1043$_{\blacktriangledown 82.4\%}$ | 0 |
| +Monitor$_{hidden}$ | | 1246$_{\blacktriangledown 78.9\%}$ | 0 |
| QwQ-32B | | 5062 | 0 |
| +Monitor$_{express}$ | 32k | 2002$_{\blacktriangledown 60.5\%}$ | 0 |
| +Monitor$_{hidden}$ | | 274$_{\blacktriangledown 94.6\%}$ | 0 |

Table 11: Monitoring strategies with maximum context length.

Under these settings, Table 11 shows that the models always produce complete answers (Overflow = 0), and our monitoring strategies still *reduce token usage significantly as expected* (Acc and HA are the same as under 2k or 4k context length).

Moreover, it is worth noting that *our proposed strategies are not constrained by context length*:

- Monitor$_{express}$ continuously tracks the densities of confident and uncertain expressions throughout the reasoning process and can detect unsolvable questions *at an early stage* (Figure 4), well before the model reaches the end of reasoning.
- Monitor$_{hidden}$ identifies unsolvable questions directly from the hidden states of last input token, i.e., *even before the model begins generating its reasoning*.

Because our strategies operate early in the reasoning trajectory, we intentionally used relatively short context length (e.g., 2k or 4k) in Section 6 to demonstrate their early detection capabilities.

# J UNPRODUCTIVE REASONING

Figure 9 and Figure 10, demonstrate representative unproductive reasoning phenomena in large language models, including repetitive looping and error accumulation, which impair their capacity for sound and consistent reasoning in complex problem-solving scenarios. They respectively show how models get stuck in repetitive deductions (Figure 9) and how errors compound during formula application and result derivation (Figure 10). In mathematical problem reasoning, the former presents repeated deduction of the same uncertain conclusion, while the latter sees continuous error accumulation due to flawed initial methods.

**Question from AIME24:**

Let $b \ge 2$ be an integer. Call a positive integer $n$ $b\text{-}\textit{eautiful}$ if it has exactly two digits when expressed in base $b$ and these two digits sum to $\sqrt{n}$. For example, $81$ is $13\text{-}\textit{eautiful}$ because $81 = \underline{6} \ \underline{3}_{13}$ and $6 + 3 = \sqrt{81}$. Find the least integer $b \ge 2$ for which there are more than ten $b\text{-}\textit{eautiful}$ integers.

**Answer from QwQ-32B:**

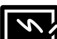 **Restating the Problem**

Okay, so I need to find the least integer b (where b is at least 2) such that there are more than ten b-beautiful numbers. A b-beautiful number is a two-digit number in base b, and the sum of its digits equals the square root of the number itself.

**? Uncertain Attempts**

....... Wait, something is **wrong** here.

......

Wait, perhaps my formula is **wrong**?

Wait, let's go back.

......

**Hmm.**
**Wait, perhaps the minimal b is 121 with 7 solutions, but the problem requires more than ten.**
**Maybe the answer is 121, but I'm not sure.**
**Alternatively, perhaps the minimal b is 256 with 8 solutions.**
**Alternatively, maybe the minimal b is 100 with 3 solutions, but the problem requires more than ten. I must be missing something.**
**Wait, perhaps I made a mistake in the earlier examples. Let me try b= 25 (k=24):**
**S from5 to24.**
**S=9:9*8=72 divisible by24? 72/24=3. Yes.**
**S=16:16*15=240 divisible by24? 240/24=10. Yes.**
**S=24:24*23=552 divisible by24? 552/24=23. Yes.**
**Thus count=3.**

**∞ Repetitive Looping**

Hmm.
Wait, perhaps the minimal b is 121 with 7 solutions, but the problem requires more than ten.
**Maybe the answer is 121, but I'm not sure.**
……
Hmm.
Wait, perhaps the minimal b is 121 with 7 solutions, but the problem requires more than ten.
**Maybe the answer is 121, but I'm not sure.**
……
Hmm.
Wait, perhaps the minimal b is 121 with 7 solutions, but the problem requires more than ten.
**Maybe the answer is 121, but I'm not sure.**
……
Hmm.
Wait, perhaps the minimal b is 121 with 7 solutions, but the problem requires more than ten.
**Maybe the answer is 121, but I'm not sure.**
……
Hmm.
Wait, perhaps the minimal b is 121 with 7 solutions, but the problem requires more than ten.
**Maybe the answer is 121, but I'm not sure.**
……

Figure 9: An example of unproductive reasoning manifested as repetitive looping in LRMs when attempting to solve a mathematical problem, where the model repeatedly cycles through the same uncertain deductions without making progress.

**Question from AIME25:**

Alex divides a disk into four quadrants with two perpendicular diameters intersecting at the center of the disk. He draws $25$ more lines segments through the disk, drawing each segment by selecting two points at random on the perimeter of the disk in different quadrants and connecting these two points. Find the expected number of regions into which these $27$ line segments divide the disk.

**Answer from DeepSeek-R1-Distill-Qwen-32B :**

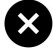 **Restating the Problem**

Okay, so I have this problem where Alex divides a disk into four quadrants using two perpendicular diameters .......

**Initial Formula**

So, the formula would be:
E[Regions] = 1 + n + E[Intersections]
Where n is the number of chords, which is 27 in this case (2 diameters + 25 chords).
......

**Incorrect Initial Formula Application (Ignoring Pre-Existing Regions)**

❌ **Error Accumulation**

......
But according to the formula, it's 1/3, which is clearly **wrong.**
Wait, this suggests that **my earlier computation is incorrect.**
......
But according to the formula, it's 1 + 2/3 $\approx$ 1.666, which is **incorrect.**
Therefore, my approach is still **flawed**.
......
Wait, let me recompute.
.......
Wait, this is getting **too confusing.**
......
Alternatively, perhaps ......
Therefore, the expected number of regions is 1 + 27 + 351*(1/9) = 1 + 27 + 39 = 67.
But in the n=3 case, this formula gives 1 + 3 + 3*(1/9) = 4 + 1/3 $\approx$ 4.333, which is **incorrect.**
......
But the problem might expect an integer, so perhaps my approach is still **incorrect.**
Wait, ........
Therefore, the formula is **incorrect.**
.......

**Forced Conclusion Despite Unresolved Inconsistencies**

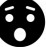 **Guessing Final Answer**

But in the n=3 case, this gives 1 + 3 + 3*(1/9) = 4 + 1/3 $\approx$ 4.333, which is **incorrect.**
Therefore, the formula is not applicable here.
I think the correct answer is 67, but **I'm not entirely sure.**
Alternatively, perhaps the expected number of regions is 1 + 27 + 39 = 67.
Therefore, the final answer is 67.
**But I'm not entirely confident  due to the inconsistencies in smaller cases.**
But **given the time** I've spent, I think the answer is 67.

Figure 10: An example of error accumulation in LRMs when tackling a problem on the expected number of regions in a disk, where the model accumulates errors due to an incorrect initial formula, leading to inconsistent results and a forced answer guess despite unresolved contradictions.

