# OpenReview forum: "On the Self-awareness of Large Reasoning Models' Capability Boundaries"
_ICLR.cc/2026/Conference — Submitted to ICLR 2026_

### Official Review · Reviewer_DRCG · 2025-10-25

**Soundness:** 2
**Presentation:** 2
**Contribution:** 3
**Rating:** 2
**Confidence:** 4

**Summary:**

This paper studies if reasoning models is able to be aware of its capability boundary. The authors provide two methods: a black-box confidence keywords detection, and a white-box hidden states classification.

**Strengths:**

- The perspective of understaing model capacity through certainty trajectory's concavity is novel;
- The authors design comprehensive experiments on multiple models and benchmarks, supporting a universality of their findings.

**Weaknesses:**

- The paper is in lack of some important details, making its phrasing not precise:
  - What is the definition of capability boundary? For example, it is confusing in Figure 3 that the capability boundary is the straight line connecting reasoning stage 0% and reasoning stage 100%.
  - How are the confident expressions and uncertain expressions defined? Although Appendix B provides a list of words, but is there any rule to construct such a list? e.g. assume a multilingual language model, how to construct the list for other languages?
  - In eq.1, how is the "model expresses more confidence/uncertainty defined"? For example, if a model solves a problems by several steps, and is only confident at the end of the reasoning process, is it considered more confident, or more uncertain?
  - In section 4, the detail how reasoning process is divided into stages is unclear.
  - What is the training set of the LDA and LR? For a given token budget, will the training set also be constrained under that budget, or is free to use as much token as possible?
- Table 3 shows the primary end-to-end experiment result for the two methods. However, it will be more supportive if the authors can provide an average answer length without any overflow constraint. As the models almost always overflow, 2K/4K-token budget may not be a reasonable choice, because the baseline method (no monitor), due to the lack of enough tokens in reasoning, could show a poor performance.
- The paper can benefit from comparing with some other confidence based early existing methods to show its speedup. Since the main metric reported is to reduce the overflow ratio, comparison with existing approaches solving overthinking is important.

**Questions:**

- The authors use Figure 3 to support a relation between expression density curve's concavity and the answer correctness However, it seems that in both correct and wrong answers, the confident expression density is always concave and the uncertain expression density is always convex.
- What is the exact number of y-axis? Is it #expression per token of the stage, or something alike?

---

> ### Author Response · Authors · 2025-11-25
> **Author response (1/5)**
>
> Dear Reviewer DRCG, thank you very much for your careful review of our paper and thoughtful comments. We are encouraged by your positive comments on the **novel perspective of understanding model capacity through certainty trajectory's concavity**, and **comprehensive experiments on multiple models and benchmarks to support a universality of their findings** of our paper. We hope the following responses can help clarify potential misunderstandings and alleviate your concerns.
>
> ---
>
> **Weakness 1**: The paper is in lack of some important details, making its phrasing not precise.
>
> **Response**: Thank you for this comment! We are sorry for not providing sufficient details in our initial submission. Below, we try to alleviate the specific issues you raised.
>
> **Issue 1: What is the definition of capability boundary? For example, it is confusing in Figure 3 that the capability boundary is the straight line connecting reasoning stage 0% and reasoning stage 100%.**
>
> In general, the capability boundary is a **criterion that can distinguish unsolvable from solvable questions at an early stage of reasoning**. In our submission, we identify this boundary from two complementary sources, reasoning expressions and hidden states of the last input token:
> - From reasoning expressions, we observe that the dominant expression trajectory is concave for solvable questions and convex for unsolvable questions. To hightlight such structual difference, we draw a straight line connecting the start and end points of the dominant trajectory in Figure 3. This makes the two types of curves clearly separable, with **solvable and unsolvable questions lying on opposite sides of the line**.
> - From hidden states, we observe that a simple linear classifier can seperate unsolvable questions from solvable ones. Therefore, in Figure 5, we **represent the capability boundary using the decision boundary learned by the linear classifier**.

---

> ### Author Response · Authors · 2025-11-25
> **Author response (2/5)**
>
> **Issue 2: How are the confident expressions and uncertain expressions defined?**
>
> We extract confident and uncertain expressions **individually for each LRM and for each task**. Therefore, for tasks in other languages, the sets of confident and uncertain expressions differ according to the same extraction pipeline. We are sorry for not making this clear in the submission. Below, we provide a more detailed description of the pipeline for identifying reasoning expressions of confidence (using GPT-oss-20B as an example):
> - 1. **Identify all anthropomorphic expressions from the model’s reasoning trace.** Prior work [1,2] show that the anthropomorphic expressions (e.g., "wait") are crucial for LRM reasoning performance, in contrast to purely neutral mathematical steps (e.g., "11 + 5 = 16"). Following this insight, we curate a set of anthropomorphic expressions that GPT-oss-20B commonly uses.
> - 2. **Select expressions whose occurrence frequency is most predictive of answer correctness.** For each question, we count the occurrences of all extracted expressions within reasoning. Using the final answer correctness as the label and the occurrence counts for each expressions as features, we apply feature-importance analysis methods [3], including SHAP and Permutation Importance, to identify the subset of expressions most informative for predicting whether the model’s final answer is correct. As shown in the following table, we select expressions such as "wait", "yes", "exactly", and "missed" which rank among the most predictive, whereas others (e.g., "so") exhibit negligible predictive value and are therefore excluded.
> - 3. **Divide the selected expressions into confidence-related and uncertainty-related categories.** This step is grounded in the connection between epistemic markers and expressed uncertainty. Prior work [4,5] show that epistemic markers reflect LLMs' internal confidence, analogous to how humans use expressions such as "I am not sure" or "it is unlikely that" to convey uncertainty. Following this insight, we classify each selected expression as reflecting confidence or uncertainty.
>
> **Table.** Feature importance score of expressions for GPT-oss-20B.
> | Expression | SHAP ↑ | Permutation Importance ↑ | Combined Score ↑ |
> |-----------|------|--------------------------|----------------|
> | wait      | 1.00 | 1.00                     | 2.00           |
> | yes       | 0.33 | 0.73                     | 1.06           |
> | exactly   | 0.26 | 0.73                     | 0.99           |
> | missed    | 0.10 | 0.86                     | 0.97           |
> | ...       |      |                          |                |
> | so        | 0.01 | 0.00                     | 0.01           |
>
> Since we extract confidence-related reasoning expressions (i.e., black-box signals) **separately for each LRM and for each task**, the effectiveness of our approach continues to hold across different settings.
>
> ---
>
> References
> - [1] DeepSeek-R1: Incentivizing Reasoning Capability in LLMs via Reinforcement Learning. Nature, 2025.
> - [2] Demystifying Reasoning Dynamics with Mutual Information: Thinking Tokens are Information Peaks in LLM Reasoning. NIPS, 2025.
> - [3] Feature Selection Strategies: a Comparative Analysis of SHAP-value and Importance-based Methods. Journal of Big Data, 2024.
> - [4] Can Large Language Models Faithfully Express Their Intrinsic Uncertainty in Words? EMNLP, 2024.
> - [5] Revisiting Epistemic Markers in Confidence Estimation: Can Markers Accurately Reflect Large Language Models' Uncertainty? ACL, 2025.

---

> ### Author Response · Authors · 2025-11-25
> **Author response (3/5)**
>
> **Issue 3: In eq.1, how is the "model expresses more confidence/uncertainty defined"? For example, if a model solves a problems by several steps, and is only confident at the end of the reasoning process, is it considered more confident, or more uncertain?**
>
> - Eq. 1 in Section 3 is **an initial toy experiment** that reflects our preliminary observation that "expressions emerge through out the reasoning process and become especially evident in its final stages..." (line 157-161). In this toy setting, "model expresses more confidence/uncertainty" is **defined using a simple matching count between confident versus uncertain expressions**.
> - In the example you described (if a model solves a problems by several steps, and is only confident at the end of the reasoning process), Eq. 1 would consider the model as "more confident" because its final reasoning stage contains a higher number of confident expressions.
> - However, this simple determination is not sufficiently rigorous. Therefore, in Section 4 we go beyond Eq. 1 and analyze how confident and uncertain expressions evolve over the entire reasoning trajectory, allowing us to more accurately characterize how these signals reflect the model’s capability boundary.
>
>
> **Issue 4: In section 4, the detail how reasoning process is divided into stages is unclear.**
>
> In Section 4, we **uniformly divide the reasoning process into 50 stages** in order to compute the densities of confident and uncertain expressions at each stage. This allows us to compare ConfDiff across stages and to compute its derivative for assessing ConfCurv.
>
> **Issue 5: What is the training set of the LDA and LR? For a given token budget, will the training set also be constrained under that budget, or is free to use as much token as possible?**
>
> - We construct a dataset by **mixing six benchmarks spanning a wide range of difficulty** (line 142: AIME...; line 266: expand our dataset by including...). We then **randomly split the dataset into an 80% training set and a 20% test set (with balanced solvable/unsolvable labels)**. We train the linear classifier (LDA and LR) on the training set, and evaluate its classification accuracy on the test set.
> - For a given token budget, **the hidden states are not constrained** by the budget because they are computed from the last input token, regardless of how long the model is allowed to generate. In contrast, **the solvable/unsolvable labels are constrained** by the budget because they depend on whether the final answer, generated under that budget, is correct. Therefore, in our experiments, **the solvable/unsolvable labels are derived from the answers without constraint on token budget**, allowing LRMs to complete their answers so that correctness can be evaluated.

---

> ### Author Response · Authors · 2025-11-25
> **Author response (4/5)**
>
> **Weakness 2**: Table 3 shows the primary end-to-end experiment result for the two methods. However, it will be more supportive if the authors can provide an average answer length without any overflow constraint. As the models almost always overflow, 2K/4K-token budget may not be a reasonable choice, because the baseline method (no monitor), due to the lack of enough tokens in reasoning, could show a poor performance.
>
> **Response**: Thank you for this insightful comment! We fully understand your concern that models with 2k/4k token budget may not be able to produce complete answers. To alleviate your concern, we provide the results (in the following table) **under extended token budget, set to the maximum** allowed by each model according to its HuggingFace model card (128k for GPT-oss-20B, 64k for R1-Distill-Qwen3-8B, and 32k for both R1-Distill-Qwen3-32B and QwQ-32B).
>
> **Table.** Results under maximum sequence length.
> | Method                     | Sequence length | Token ↓ | Overflow ↓ |
> |----------------------------|------------------|-------|----------|
> | GPT-oss-20B                | 128k             | 6023  | 0        |
> | + Monitor$\_{\text{express}}$        | 128k             | 1897  | 0        |
> | + Monitor$\_{\text{hidden}}$         | 128k             | 3826  | 0        |
> | R1-Distill-Qwen3-8B        | 64k              | 5096  | 0        |
> | + Monitor$\_{\text{express}}$        | 64k              | 2834  | 0        |
> | + Monitor$\_{\text{hidden}}$         | 64k              | 2227  | 0        |
> | R1-Distill-Qwen3-32B       | 32k              | 5910  | 0        |
> | + Monitor$\_{\text{express}}$        | 32k              | 1043  | 0        |
> | + Monitor$\_{\text{hidden}}$         | 32k              | 1246  | 0        |
> | QwQ-32B                    | 32k              | 5062  | 0        |
> | + Monitor$\_{\text{express}}$        | 32k              | 2002  | 0        |
> | + Monitor$\_{\text{hidden}}$         | 32k              | 274   | 0        |
>
> Under these settings, the models always produce complete answers (Overflow = 0), and our monitoring strategies still **reduce token usage significantly as expected** (Acc and HA are the same as under 2k or 4k token budget).
>
> Moreover, it is worth noting that **our proposed monitoring strategies are not constrained by token budget**:
> - Monitor$\_{\text{express}}$ continuously tracks the densities of confident and uncertain expressions throughout the reasoning process and can detect unsolvable questions **at an early stage** (Figure 4), well before the model reaches the end of reasoning.
> - Monitor$\_{\text{hidden}}$ identifies unsolvable questions directly from the hidden states of last input token, i.e., **even before the model begins generating its reasoning**.
>
> Because our monitoring strategies operate early in the reasoning trajectory, **we intentionally used relatively short sequence length limits (e.g., 2k or 4k) in the initial submission to demonstrate their early detection capabilities**.

---

> ### Author Response · Authors · 2025-11-25
> **Author response (5/5)**
>
> **Weakness 3**: The paper can benefit from comparing with some other confidence based early existing methods to show its speedup. Since the main metric reported is to reduce the overflow ratio, comparison with existing approaches solving overthinking is important.
>
> **Response**: Thank you for this insightful comment! To alleviate your concern, we compare our methods with a recent approach for improving reasoning efficiency through **Dynamic-Early-Exit** [1]. This method monitors LRMs' reasoning transitions and induce trial answer, which will be used for confidence evaluation. When the confidence is sufficiently high, LRMs are set to stop and generate answer.
>
> We provide the results alongside our Monitor$\_{\text{express}}$ and Monitor$\_{\text{hidden}}$ in the following table. To evaluate the "overflow ratio" you mentioned, we keep the token budget to 4k. **Although Dynamic-Early-Exit reduces token usage and overflow to some extent, it also substantially lowers overall accuracy and fails to detect unsolvable questions (HA = 0).** In contrast, our approaches enable LRMs to reliably identify unsolvable questions without compromising their accuracy on solvable ones, demonstrating **a fundamentally different capability from early-exit methods that target overthinking rather than unsolvability**.
>
> **Table.** Dynamic-Early-Exit compared to our monitoring strategies.
> | Method                   | Acc ↑ | HA ↑   | Token ↓ | Overflow ↓ |
> |--------------------------|-------|--------|----------|-------------|
> | GPT-oss-20B              | 74.6% | 0      | 4015     | 97%         |
> | + Dynamic-Early-Exit     | 49.8% | 0      | 3643     | 19.4%       |
> | + Monitor$\_{\text{express}}$       | 74.6% | 100%   | 1499     | 15.2%       |
> | + Monitor$\_{\text{hidden}}$       | 73.7% | 98.9%  | 3317     | 5.4%        |
>
>
> This advantage arises because our scenario is fundamentally different from overthinking:
> - The prior work to mitigate overthinking [1-3] **focus on cases where the model can solve the question** but wastes computation by reasoning for too long. These methods do not attempt to identify questions that lie beyond the model’s capability boundary. Therefore, **they do not acknowledge unsolvable questions and continue producing unproductive reasoning for these questions**.
> - In contrast, our work **targets a complementary and less-studied scenario: when a question lies beyond the model’s capability boundary, LRMs continue to generate lengthy but unproductive reasoning despite having no chance of arriving at a correct answer**. Our goal is to detect such cases early, even before reasoning begins.
>
> ---
>
> References
> - [1] Dynamic Early Exit in Reasoning Models. Arxiv, 2025.
> - [2] Stop Overthinking: A Survey on Efficient Reasoning for Large Language Models. TMLR, 2025.
> - [3] Do NOT Think That Much for 2+ 3=? On the Overthinking of Long Reasoning Models. ICML, 2025.
>
> ---
>
> **Question 1**: The authors use Figure 3 to support a relation between expression density curve's concavity and the answer correctness. However, it seems that in both correct and wrong answers, the confident expression density is always concave and the uncertain expression density is always convex.
>
> **Response**: Thank you for this question! We are sorry that there might be a potential misunderstanding. In Figure 3, the concavity and convexity are computed on the **dominant expression trajectory**. So for solvable questions, the dominant trajectory is concave, whereas for unsolvable questions, the dominant trajectory is convex.
>
> ---
>
> **Question 2**: What is the exact number of y-axis? Is it #expression per token of the stage, or something alike?
>
> **Response**: Thank you for this question! The exact number of y-axis is the **proportion of tokens within a given stage $t$ that correspond to confident or uncertain expressions**:
>
> $
> \mathcal{D}_C(t) = \frac{\text{Number of tokens corresponding to confident expressions}}{\text{Total number of tokens within stage t}}
> $
>
> $
> \mathcal{D}_U(t) = \frac{\text{Number of tokens corresponding to uncertain expressions}}{\text{Total number of tokens within stage t}}
> $

---

> ### Comment · Reviewer_DRCG · 2025-11-26
> **Feedback to the authors' response**
>
> Greetings the authors,
>
> Thank you for your response to my concerns. Below is the feedback:
>
> 1.1 and Q1 The explanation to the hidden state based capability boundary is clear to me. Regarding the resoning expression based method, as the response in Q1 mentioned, the statement is reduced to "confident expressions dominates in correct answers, and uncertain expressions dominates in wrong answers", because confident expressions' trajectory is always concave, while uncertain's is always convex. Although it does not hurt the primary result and motivation for ConfDiff, this challenges the motivation of ConfCurv.
>
> 1.2 The process to get the expression is clear to me, although the "per-model, per-task" paradigm is slightly cost inefficient.
>
> 1.3 I have no more question on this, thanks for your clarification
>
> 1.4 When verifying on a known response, it is possible to evenly divide the whole response into 50 stages. However, when tracing a generating response, how is the stage defined? i.e., how is the "t" and "s" in expression (2)(3) defined during generation?
>
> 1.5 I have no more question on this, thanks for your clarification
>
> 2 Thanks for providing the additional result. My concern is addressed.
>
> 3 Thanks for providing the end-to-end comparison with the new early exit based baseline. It is surprising to see that Dynamic-Early-Exit performs poorly on the model accuracy, which is different from the number they reported in the paper. Since in the Dynamic-Early-Exit paper, the performance is also evaluated on GSM8K, AMC23 AMC24, etc, is there an explanation to the poor performance when simply migrating to another LRM but on the same dataset?
>
> I've accordingly updated my scoring.

---

> > ### Author Response · Authors · 2025-11-27
> > **Author response to feedback**
> >
> > Dear Reviewer DRCG, thank you very much for your thoughtful feedback and support for our work! We are encouraged that several of your concerns have been alleviated, and we appreciate the opportunity to clarify the remaining points.
> >
> > ---
> >
> > **1.1 and Q1: Although it does not hurt the primary result and motivation for ConfDiff, this challenges the motivation of ConfCurv.**
> >
> > Thank you for this comment! We fully agree with you that, compared to ConfDiff, it might not be necessary to design ConfCurv. However, our study follows a "**phenomenon → hypothesis → validation**" paradigm.
> >
> > We begin by observing two types of capability boundary signals (line 211-215) in Figure 3:
> > 1. the vertical ordering of the confident and uncertain trajectories
> > 2. the convexity/concavity of dominant trajectory
> >
> > Motivated by these observations, we define corresponding indicators, ConfDiff (for vertical ordering) and ConfCurv (for convexity/concavity), to evaluate whether these signals indeed reflect unsolvable/solvable questions. The experimental results in Figure 4 confirm that both ConfDiff and ConfCurv capture capability boundary signals, **although ConfDiff proves to be more reliable (line 250)**.
> >
> > Therefore, we think it is worth trying to use ConfCurv to study the convexity/concavity signal. Based on experimental results in Figure 4, we prefer to use ConfDiff as the primary and more reliable indicator in our subsequent analyses.
> >
> > ---
> >
> > **1.2: The "per-model, per-task" paradigm is slightly cost inefficient.**
> >
> > Thank you for this comment! We fully agree with you that extracting confident and uncertain expressions "per-model, per-task" might be less efficient than preparing one unified list. However, once the complete answers and correctness labels are available, our extraction pipeline incurs **minimal overhead** (~ 10s), due to the efficiency of the feature-importance analysis methods SHAP and Permutation Importance.
> >
> > In contrast, processing a single unsolvable question with H20 on GPT-oss-20B takes on average ~5 minutes. Thus, the cost of preparing model- and task-specific expression lists is negligible compared to the overall computational cost.
> >
> > Moreover, because different models exhibit different capability boundaries across different tasks, it is more rigorous and reliable to constructing expression lists per-model and per-task.
> >
> > ---
> >
> > **1.4: When verifying on a known response, it is possible to evenly divide the whole response into 50 stages. However, when tracing a generating response, how is the stage defined? i.e., how is the "t" and "s" in expression (2)(3) defined during generation?**
> >
> > Thank you for this question! We are sorry for not making this sufficiently clear in our submission. Section 4 analyzes capabiltiy boundary from known response with complete reasoning trace, where we divide the reasoning into 50 uniform stages. From this analysis, we observe that boundary awareness emerges at an early stage (e.g., 2% reasoning stage for GPT-oss-20B). This means that, for a complete answer consisting of 6000 tokens for example, **comparing the density between confident and uncertain expressions at only $6000 \times 2 \\% = 120$ tokens is already predictive of the final answer's correctness.**
> >
> > Therefore, during generation when the full sequence is not yet available, we operationalize the "stage" by **monitoring the reasoning after the first 120 generated tokens (following the 2% ratio discovered in the full-trace analysis)**. We compute the densities of confident and uncertain expressions. If uncertain expressions dominate, indicating an unsolvable question, we early stop the reasoning and reprompt the model with the self-awareness prompt suffix (line 340-349).
> >
> > ---
> >
> > **3: In the Dynamic-Early-Exit paper, the performance is also evaluated on GSM8K, AMC23 AMC24, etc, is there an explanation to the poor performance when simply migrating to another LRM but on the same dataset?**
> >
> > Thank you for this question! Our evaluation **includes two additional datasets, HLE and HMMT**, which are not evaluated in the Dynamic-Early-Exit paper. We also observe that many of the failure cases (Dynamic-Early-Exit turns question the model would have answered correctly into incorrect ones) come primarily from HLE and HMMT, containing **hard but still solvable questions**.
> >
> > Dynamic-Early-Exit triggers an exit when the model's trial answer (elicited by "final answer") reaches sufficiently high confidence (defined as the maximum predicted probability). A potential explanation for its poor performance might be: for these harder but solvable questions, **the model's confidence may be overly biased by the phrase "final answer".** This can cause wrong exits when the model has not reach the correct answer.
> >
> > This is also aligned with why, in our work, **we do not include "final answer" in the expression list**.

---

### Official Review · Reviewer_haJD · 2025-10-28

**Soundness:** 3
**Presentation:** 3
**Contribution:** 3
**Rating:** 6
**Confidence:** 2

**Summary:**

This paper claims that LRMs can excel at math but often overthink hard questions, wasting tokens and still getting them wrong.
This paper tests whether models know their own limits: in black-box use, confidence/uncertainty trajectories in the reasoning text signal solvable vs. unsolvable cases; in white-box use, the last-input hidden state linearly separates them before any reasoning.
Leveraging these signals, the authors monitor reasoning expressions or hidden states to stop unproductive reasoning early.
They report large token savings (62.7–93.6%) without hurting accuracy.

**Strengths:**

- The paper frames "capability boundaries" clearly and turns that idea into two simple, test-time strategies that are easy to implement.

- Evidence comes from both black-box signals (confidence/uncertainty trajectories) and white-box signals (last-token hidden states), which makes the story feel cohesive.

- Reported token savings are large without hurting accuracy, suggesting real efficiency wins for math-style workloads.

**Weaknesses:**

- The striking linear separability in hidden states may reflect dataset or difficulty splits rather than genuine boundary awareness, so the claim risks a distribution confound.

- The black-box indicators depend on specific reasoning styles and cue phrases, which may not hold under different prompts, languages, or chain-of-thought formats.

- The evaluation is narrow and misses stronger uncertainty/abstention baselines and cross-model tests, and the white-box path raises deployability concerns where internal states are inaccessible.

**Questions:**

- Can you show the hidden-state separability within a single dataset and difficulty-controlled splits, with train and test drawn from the same distribution, and report how the accuracy/AUC changes?

- How sensitive are the black-box signals to prompt style, language, and removal of common cue phrases, and do the effects hold under alternative chain-of-thought formats or shorter reasoning budgets?

- Under matched compute, how do your methods compare to stronger uncertainty and verifier-guided strategies, and can the black-box-only variant deliver similar gains when white-box access to hidden states is unavailable?

---

> ### Author Response · Authors · 2025-11-25
> **Author response (1/3)**
>
> Dear Reviewer haJD, thank you very much for your careful review of our paper and thoughtful comments. We are encouraged by your positive comments on the **clear frame of "capability boundaries"**, **easy to implement test-time strategies**, **cohesive evidence**, and **real efficiency for math-style workloads without hurting accuracy** of our paper. We hope the following responses can help clarify potential misunderstandings and alleviate your concerns.
>
> ---
>
> **Weakness 1 and Question 1**: The striking linear separability in hidden states may reflect dataset or difficulty splits rather than genuine boundary awareness, so the claim risks a distribution confound. Can you show the hidden-state separability within a single dataset and difficulty-controlled splits, with train and test drawn from the same distribution, and report how the accuracy/AUC changes?
>
> **Response**: Thank you for this insightful comment! We fully agree with your concern that the linear separability may reflect difficult splits rather than genuine boundary awareness. However, there might be a potential misunderstanding: **the ability to recognize questions from different difficulty splits is not in conflict with a model’s boundary awareness**.
>
> To illustrate, consider humans with self-awareness of capability boundary. In an exam with limited time and many hard problems, skilled test-takers are able to quickly identify questions beyond their capability and avoid wasting time on them. They may recognize that the question resembles ones they consistently failed to solve during preparation (analogous to our Monitor$\_{\text{hidden}}$ strategy), or they may attempt a few initial reasoning steps and realize they lack confidence to proceed (analogous to our Monitor$\_{\text{express}}$ strategy). **Such self-awareness arises precisely because humans have previously encountered hard problems; similarly, the hard questions in HLE play this role for LRMs.**
>
> Still, we find your suggestion on "show the hidden-state separability within a single dataset and difficulty-controlled splits" extremely valuable. We provide in the following table the accuracy of separating unsolvable questions from solvable ones **within each benchmark using LDA and LR classifiers trained and tested on splits from the same dataset**. Specifically, for both HLE and GSM8K, we randomly split each dataset into an 80% training set and a 20% test set (with balanced solvable/unsolvable labels), train the linear classifier on the training set, and evaluate its classification accuracy on the test set.
>
> **Table.** Accuracy (%) of separating solvable and unsolvable questions in hidden states using linear classifiers trained and tested on subsets split within the same benchmark.
> | Dataset | Model                  | LDA  | LR   |
> |---------|-------------------------|------|------|
> | HLE     | GPT-oss-20B            | 89.3 | 90.3 |
> | HLE     | R1-Distill-Qwen3-8B    | 94.9 | 96.4 |
> | HLE     | R1-Distill-Qwen3-32B   | 96.4 | 96.4 |
> | HLE     | QwQ-32B                | 93.4 | 96.4 |
> | GSM8k   | GPT-oss-20B            | 80.3 | 83.3 |
> | GSM8k   | R1-Distill-Qwen3-8B    | 70.1 | 71.6 |
> | GSM8k   | R1-Distill-Qwen3-32B   | 85.2 | 89.0 |
> | GSM8k   | QwQ-32B                | 87.9 | 87.1 |
>
> The results show that **linear classifiers can still separate solvable and unsolvable questions from hidden states, although with reduced accuracy compared to aggregated benchmarks setting (Table 2)**. This drop is small on HLE but larger on GSM8K. The reason is that **HLE contains a larger proportion of unsolvable questions, enabling the linear classifiers to better learn the hidden state information associated with unsolvability**. Consequently, these unsolvable questions in HLE can be effectively leveraged to develop models' Hard Abstention capability.

---

> ### Author Response · Authors · 2025-11-25
> **Author response (2/3)**
>
> **Weakness 2 and Question 2**: The black-box indicators depend on specific reasoning styles and cue phrases, which may not hold under different prompts, languages, or chain-of-thought formats. How sensitive are the black-box signals to prompt style, language, and removal of common cue phrases, and do the effects hold under alternative chain-of-thought formats or shorter reasoning budgets?
>
> **Response**: Thank you for this insightful comment! We fully agree with you that the black-box indicators depend on specific reasoning styles and cue phrases. This is precisely why we extract confident and uncertain expressions **individually for each LRM based on its reasoning style**. We are sorry for not making this clear in the submission. Below, we provide a more detailed description of the pipeline for identifying reasoning expressions of confidence (using GPT-oss-20B as an example):
> - 1. **Identify all anthropomorphic expressions from the model’s reasoning trace.** Prior work [1,2] show that the anthropomorphic expressions (e.g., "wait") are crucial for LRM reasoning performance, in contrast to purely neutral mathematical steps (e.g., "11 + 5 = 16"). Following this insight, we curate a set of anthropomorphic expressions that GPT-oss-20B commonly uses.
> - 2. **Select expressions whose occurrence frequency is most predictive of answer correctness.** For each question, we count the occurrences of all extracted expressions within reasoning. Using the final answer correctness as the label and the occurrence counts for each expressions as features, we apply feature-importance analysis methods [3], including SHAP and Permutation Importance, to identify the subset of expressions most informative for predicting whether the model’s final answer is correct. As shown in the following table, we select expressions such as "wait", "yes", "exactly", and "missed" which rank among the most predictive, whereas others (e.g., "so") exhibit negligible predictive value and are therefore excluded.
> - 3. **Divide the selected expressions into confidence-related and uncertainty-related categories.** This step is grounded in the connection between epistemic markers and expressed uncertainty. Prior work [4,5] show that epistemic markers reflect LLMs' internal confidence, analogous to how humans use expressions such as "I am not sure" or "it is unlikely that" to convey uncertainty. Following this insight, we classify each selected expression as reflecting confidence or uncertainty.
>
> **Table.** Feature importance score of expressions for GPT-oss-20B.
> | Expression | SHAP ↑ | Permutation Importance ↑ | Combined Score ↑ |
> |-----------|------|--------------------------|----------------|
> | wait      | 1.00 | 1.00                     | 2.00           |
> | yes       | 0.33 | 0.73                     | 1.06           |
> | exactly   | 0.26 | 0.73                     | 0.99           |
> | missed    | 0.10 | 0.86                     | 0.97           |
> | ...       |      |                          |                |
> | so        | 0.01 | 0.00                     | 0.01           |
>
> Since we extract confidence-related reasoning expressions (i.e., black-box signals) **separately for each LRM and for each task**, the effectiveness of our approach continues to hold across different settings.
>
> ---
>
> References
> - [1] DeepSeek-R1: Incentivizing Reasoning Capability in LLMs via Reinforcement Learning. Nature, 2025.
> - [2] Demystifying Reasoning Dynamics with Mutual Information: Thinking Tokens are Information Peaks in LLM Reasoning. NIPS, 2025.
> - [3] Feature Selection Strategies: a Comparative Analysis of SHAP-value and Importance-based Methods. Journal of Big Data, 2024.
> - [4] Can Large Language Models Faithfully Express Their Intrinsic Uncertainty in Words? EMNLP, 2024.
> - [5] Revisiting Epistemic Markers in Confidence Estimation: Can Markers Accurately Reflect Large Language Models' Uncertainty? ACL, 2025.

---

> ### Author Response · Authors · 2025-11-25
> **Author response (3/3)**
>
> **Weakness 3 and Question 3**: The evaluation is narrow and misses stronger uncertainty/abstention baselines and cross-model tests, and the white-box path raises deployability concerns where internal states are inaccessible. Under matched compute, how do your methods compare to stronger uncertainty and verifier-guided strategies, and can the black-box-only variant deliver similar gains when white-box access to hidden states is unavailable?
>
> **Response**: Thank you for this comment! We fully understand your concern that the evaluation can be extended. However, there might be potential misunderstanding.
>
> **Regarding cross-model tests**:
> Each LRM exhibits different capabilities, and therefore their **capability boundaries are inherently model-specific**. For this reason, the study of capability boundaries should be conducted **within each model and each task**, rather than across different models. Cross-model tests are not particularly meaningful, for example, the capability boundary of GPT-5 does not generalize to GPT-3.
>
> **Regarding deployability concerns of white-box path:**
> We fully agree with you that Monitor$\_{\text{hidden}}$ (white-box path) requires access to hidden states. This is precisely why we also introduce a black-box path, Monitor$\_{\text{express}}$, to complement the white-box approach. Notably, **the black-box path achieves comparable gains, and in some cases even outperforms the white-box path** (e.g., GPT-oss-20B in Table 3 of our submission), demonstrating its practical deployability when hidden states access is unavailable.
>
> **Regarding uncertainty/abstention baselines:**
> As suggested, we provide comparisons to uncertainty and verifier-guided strategies: model-as-judge approach [1] and uncertainty-estimation approach [2,3]. However, our work study **abstention on unsolvable questions**, and therefore these methods cannot directly serve as a baseline for our setting. Consequently, we first clarify how we adapt these methods to fit our task.
>
> **Model-as-judge:** For each question, we ask the model to evaluate whether its own predicted answer is correct using the following prompt template:
> > Question: {question}
> Proposed Answer: {answer}
> Is the proposed answer:
> A True
> B False
> The proposed answer is:
>
> This follows prior work [1] which study whether language models can evaluate the validity of their own claims by asking models to first propose answers, and then to evaluate that their answers are correct. If the model judges its own incorrect answer to be incorrect, we count this as a hard abstention (HA). Conversely, if the model judges its correct answer to be incorrect, this reduces the overall task accuracy (Acc). Formally,
>
> $
> \text{Acc} = \mathbb{P} \left(
>     \text{Model judges the answer to be correct} \land
>     \text{Answer is correct}
> \right)
> $
>
> $
> \text{HA} = \mathbb{P} \left(
>     \text{Model judges the answer to be incorrect}
>     \mid
>     \text{Answer is incorrect}
> \right)
> $
>
> **Uncertainty-estimation:** For each question, we compute the semantic entropy of the model’s generated answer, where a higher entropy indicates greater uncertainty. This follows prior work [2,3], which introduces semantic entropy as an uncertainty measure that captures linguistic invariances induced by shared meanings. When the entropy exceeds a certain threshold, the model is considered uncertain about its answer. Similar as before, we define:
>
> $
> \text{Acc} = \mathbb{P} \left(
>     \text{Semantic entropy of the answer < threshold} \land
>     \text{Answer is correct}
> \right)
> $
>
> $
> \text{HA} = \mathbb{P} \left(
>     \text{Semantic entropy of the answer >= threshold}
>     \mid
>     \text{Answer is incorrect}
> \right)
> $
>
>
> **Results:** We provide the results of these two baselines alongside our Monitor$\_{\text{express}}$ and Monitor$\_{\text{hidden}}$ in the following table. **Although both baselines provide models with a certain degree of hard abstention capability, they also incorrectly classify many solvable questions as unsolvable, which significantly reduces overall accuracy.** In contrast, our approaches enable LRMs to reliably identify unsolvable questions without compromising their accuracy on solvable ones.
>
> **Table.** Accuracy (Acc) and Hard Abstention (HA) of two baselines alongside our monitoring strategies.
> | Method                 | Acc ↑  | HA ↑|
> |------------------------|--------|--------|
> | GPT-oss-20B            | 74.6%  | 0      |
> | + Model-as-judge       | 32.1%  | 46.0%  |
> | + Uncertainty-estimation | 50.0%  | 29.0%     |
> | + Monitor$\_{\text{express}}$    | 74.6%  | 100%   |
> | + Monitor$\_{\text{hidden}}$     | 73.7%  | 98.9%  |
>
> ---
>
> References
> - [1] Language Models (Mostly) Know What They Know. Anthropic, 2022.
> - [2] Semantic Uncertainty: Linguistic Invariances for Uncertainty Estimation in Natural Language Generation. ICLR, 2023.
> - [3] Detecting Hallucinations in Large Language Models Using Semantic Entropy. Nature, 2024.

---

> ### Author Response · Authors · 2025-11-28
> **Thanks to Reviewer haJD**
>
> Please allow us to thank you again for reviewing our paper and the valuable feedback, and in particular for recognizing the strengths of our paper in terms of **clear frame of "capability boundaries", easy to implement test-time strategies, cohesive evidence, and real efficiency for math-style workloads without hurting accuracy**.
>
> Kindly let us know if our response and the new experiments have properly addressed your concerns. We are more than happy to answer any additional questions during the rebuttal period. Your feedback will be greatly appreciated.

---

### Official Review · Reviewer_93Mj · 2025-10-31

**Soundness:** 3
**Presentation:** 3
**Contribution:** 2
**Rating:** 4
**Confidence:** 4

**Summary:**

This paper mainly focuses on a widely acknowledged problem: whether LLMs know they do not know. Specifically, it proposes solutions for both black-box LLMs and open-source LLMs. Downstream applications indicate that the proposed approach can improve the LRM efficiency without sacrificing accuracy.

**Strengths:**

First, this paper addresses an important research problem whose significance is widely recognized. Its biggest strength in my opinion is the clarity of presentation along with supporting results. I would recommend it to anyone interested in understanding the current state of research on this topic.

**Weaknesses:**

I am concerned that this paper does not offer substantial additional value beyond several existing works. For example, [1] (and lots of other work including some surveys) presents similar approaches to probing the hidden states of LRMs, also supported by strong empirical results. For this reason, although I like the paper, I do not believe we need to have another paper with similar contributions.

In addition, I think it would be interesting to focus on scenarios where LLM has reasonable accuracy. As the paper mentioned, GSM8K is too easy, and HLE is too difficult. It's not that surprising LLMs have "self-awareness".


[1] Reasoning Models Know When They're Right: Probing Hidden States for Self-Verification COLM 2025

**Questions:**

N/A

---

> ### Author Response · Authors · 2025-11-25
> **Author response (1/2)**
>
> Dear Reviewer 93Mj, thank you very much for your careful review of our paper and thoughtful comments. We are encouraged by your positive comments on the **important research problem whose significance is widely recognized**, **clarity of presentation along with supporting results**, and **I would recommend it to anyone interested** of our paper. We hope the following responses can help clarify potential misunderstandings and alleviate your concerns.
>
> ---
>
> **Weakness 1**: I am concerned that this paper does not offer substantial additional value beyond several existing works. For example, [1] (and lots of other work including some surveys) presents similar approaches to probing the hidden states of LRMs, also supported by strong empirical results. For this reason, although I like the paper, I do not believe we need to have another paper with similar contributions.
>
> **Response**: Thank you for this comment! We fully understand your concern that [1] also probes the hidden states of LRMs. Indeed, this is why we cited this work in our submission (line 119). However, there might be potential misunderstanding regarding the similarity between our work and prior work:
> - [1] along with other related work [2,3] **focus on cases where the model can solve the question** but wastes computation by reasoning for too long. These methods do not attempt to identify questions that lie beyond the model’s capability boundary. Therefore, **they do not acknowledge unsolvable questions and continue producing unproductive reasoning for these questions**.
> - In contrast, our work **targets a complementary and less-studied scenario: when a question lies beyond the model’s capability boundary, LRMs continue to generate lengthy but unproductive reasoning despite having no chance of arriving at a correct answer**. Our goal is to detect such cases early, even before reasoning begins.
>
> These represent two fundamentally different settings, and the methods are not transferable:
> - For example, [1] implements a confidence-based early-exit strategy, which exits early only when an intermediate answer reaches high confidence. **This works for solvable questions**. However, when LRMs face an unsolvable question, the intermediate answers might remain low confidence throughout the entire reasoning process (Figure 9 in Appendix E). Consequently, **early-exit methods never trigger, resulting in unproductive reasoning and no hope of producing a correct answer**.
> - In contrast, our approach can detect unsolvability early, even directly from the hidden states of the last input token (Monitor$\_{\text{hidden}}$) and **explicitly acknowledge that the model cannot solve the problem, while providing a concise potential solution approach instead**.
>
> ---
>
> References
> - [1] Reasoning Models Know When They're Right: Probing Hidden States for Self-Verification. COLM, 2025.
> - [2] Deep Think with Confidence. Arxiv, 2025.
> - [3] Efficiently Scaling LLM Reasoning with Certaindex. NIPS, 2025.

---

> ### Author Response · Authors · 2025-11-25
> **Author response (2/2)**
>
> **Weakness 2**: In addition, I think it would be interesting to focus on scenarios where LLM has reasonable accuracy. As the paper mentioned, GSM8K is too easy, and HLE is too difficult. It's not that surprising LLMs have "self-awareness".
>
> **Response**: Thank you for this insightful comment! We fully agree with you that it would be interesting to focus on scenarios where LLM has reasonable accuracy. This is precisely why we construct a dataset by **mixing six benchmarks spanning a wide range of difficulty**(line 142: AIME...; line 266: expand our dataset by including...), resulting in LRMs achieving around 50% accuracy overall.
>
> Also, there might be a potential misunderstanding regarding the statement "not surprising to have self-awareness": **constructing a dataset including easy GSM8k and difficult HLE is not in conflict with a model’s self-awareness of capability boundary**.
>
> To illustrate this point, consider humans with clear self-awareness of their capability boundary. In an exam with limited time and many hard problems, skilled test-takers are able to quickly identify questions beyond their capability and avoid wasting time on them. They may recognize that the question resembles ones they consistently failed to solve during preparation (analogous to our Monitor$\_{\text{hidden}}$ strategy), or they may attempt a few initial reasoning steps and realize they lack confidence to proceed (analogous to our Monitor$\_{\text{express}}$ strategy). **Such self-awareness arises precisely because humans have previously encountered hard problems; similarly, the hard questions in HLE play this role for LRMs.**
>
> To further validate this intuition from human exam, we conduct a **cross-dataset generalization test**: we learn the capability boundary from the hidden states on HLE (i.e., training a linear classifier), and evaluate whether this boundary generalizes to GSM8k and AIME.
>
> Because **HLE contains a larger proportion of unsolvable questions, the linear classifiers can more effectively learn hidden state patterns associated with unsolvability.** Consequently, HLE’s unsolvable questions serve as a strong source of supervision for developing Hard Abstention (HA) capability.
>
>
> Therefore, we train linear classifiers on HLE and evaluate models' Hard Abstention capability on GSM8k and AIME to test whether this capability generalizes across benchmarks. As shown in the following table, **linear classifiers trained on HLE are able to identify nearly all unsolvable questions in GSM8K and AIME**, achieving HA close to 100%.
>
> **Table.** Hard Abstention (%) of linear classifier trained on HLE and test on GSM8k and AIME.
> | Model                 | GSM8k | AIME |
> |-----------------------|--------|------|
> | GPT-oss-20B           | 0      | 0    |
> | + Monitor$\_{\text{hidden}}$    | 100    | 100  |
> | R1-Distill-Qwen3-8B   | 0      | 0    |
> | + Monitor$\_{\text{hidden}}$    | 100    | 100  |
> | R1-Distill-Qwen3-32B  | 0      | 0    |
> | + Monitor$\_{\text{hidden}}$    | 90.5   | 94.7 |
> | QwQ-32B               | 0      | 0    |
> | + Monitor$\_{\text{hidden}}$    | 98.9   | 100  |
>
> In contrast, training on GSM8K is insufficient for developing Hard Abstention capability, because GSM8K contains very few unsolvable questions. The linear classifiers cannot reliably learn the hidden state patterns associated with unsolvability.

---

> ### Author Response · Authors · 2025-11-28
> **Thanks to Reviewer 93Mj**
>
> Please allow us to thank you again for reviewing our paper and the valuable feedback, and in particular for recognizing the strengths of our paper in terms of **important research problem whose significance is widely recognized, clarity of presentation along with supporting results, and I would recommend it to anyone interested**.
>
> Kindly let us know if our response and the new experiments have properly addressed your concerns. We are more than happy to answer any additional questions during the rebuttal period. Your feedback will be greatly appreciated.

---

### Official Review · Reviewer_c4xf · 2025-11-01

**Soundness:** 3
**Presentation:** 3
**Contribution:** 3
**Rating:** 6
**Confidence:** 4

**Summary:**

The paper first study the effect that LRM is self-aware of its capability boundary at reasoning process. The paper then propose two methods, reasoning expression monitoring and hidden states monitoring, to monitor black-box and white-box LLM's reasoning trajectory to determine if the model has hit its capability boundary for the given task. The evaluation shows impressive result of the two proposed methods to reduce token usage for hard (impossible) questions.

**Strengths:**

1. Novelty. The analysis and methodology is novel and interesting.
2. The paper writing is clear and concise. The evidence is presented very cleanly, and the logic flows very naturally.
3. Methodology and experiment. The methodology proposed is sound, intuitive, effective in practice, and backed by a lot of experiments and ablations to show its effectiveness.

**Weaknesses:**

Related works. Entropy-based early-exit methods (especially training-free methods) has been quite extensively studied in prior works such as the following. The paper may want to expand a bit more to make a clearer distinction between the proposed methodology and the prior works (in the related work). Just listing a few as addition to the current related work section:

- https://arxiv.org/abs/2504.01296
- https://arxiv.org/abs/2508.15260
- https://arxiv.org/abs/2412.20993
- https://arxiv.org/abs/2412.18547
- https://arxiv.org/abs/2207.05221

**Questions:**

1. Generalizability on coding tasks. The experiment mostly focuses on multiple choice questions or questions with numerical answers. Does it generalizes to coding questions? How would the methodology adapt to coding questions (and possibly more open-ended questions)?
2. Ablation on the self-aware output prefix. Does changing the output prefix alter the result?

---

> ### Author Response · Authors · 2025-11-25
> **Author response (1/3)**
>
> Dear Reviewer c4xf, thank you very much for your careful review of our paper and thoughtful comments. We are encouraged by your positive comments on the **novelty**, **clear and concise paper writing**, **natural logic flow**, and **sound, intuitive, effective in practice methodology and experiment** of our work. We hope the following responses can help clarify potential misunderstandings and alleviate your concerns.
>
> ---
>
> **Weakness 1**: The paper may want to expand a bit more to make a clearer distinction between the proposed methodology and the prior works (in the related work).
>
> **Response**: Thank you for this insightful comment and for pointing us to additional related work [1-5]! We fully agree with you that entropy-based early-exit methods has been quite extensively studied in prior works. However, there might be potential misunderstanding regarding the similarity between our work and prior work:
> - These prior work **focus on efficient reasoning in cases where the model can solve the question** but wastes computation by reasoning for too long. These methods do not attempt to identify questions that lie beyond the model’s capability boundary. Therefore, **they do not acknowledge unsolvable questions and continue producing unproductive reasoning for these questions**.
> - In contrast, our work **targets a complementary and less-studied scenario: when a question lies beyond the model’s capability boundary, LRMs continue to generate lengthy but unproductive reasoning despite having no chance of arriving at a correct answer**. Our goal is to detect such cases early, even before reasoning begins.
>
> In terms of methodology, our Monitor$\_{\text{express}}$ and Monitor$\_{\text{hidden}}$ also differ from the works you provided:
> - [1] trains the long-thinking LLMs via reinforcement learning (RL) with an added token limit, whereas our monitoring strategies are entirely **training-free**.
> - [2,3] leverages model internal confidence signals or certain point during reasoning to filter out unnecessary reasoning trace, while Monitor$\_{\text{hidden}}$ **detects unsolvability much earlier**, even from the hidden states of the last input token, before any reasoning begins.
> - [4] compresses unnecessary reasoning by including a reasonable token budget in the prompt. **Such design within prompt resembles [6], which we include as a baseline** in our submission and show its limitations in detecting unsolvable questions and reducing token usage (Table 3).
> - [5] asks models to first propose answers, and then to evalute whether their answers are correct. Our strategies evalute whether their answers are correct during reasoing (Monitor$\_{\text{express}}$) and before reasoning even starts (Monitor$\_{\text{hidden}}$), making them strictly more efficient because **no additional inference is needed**.
>
> We will revise the Related Work section to cite these prior works and more clearly distinguish our contributions from them.
>
> ---
>
> References
> - [1] ThinkPrune: Pruning Long Chain-of-Thought of LLMs via Reinforcement Learning. Arxiv, 2025.
> - [2] Deep Think with Confidence. Arxiv, 2025.
> - [3] Efficiently Scaling LLM Reasoning with Certaindex. NIPS, 2025.
> - [4] Token-Budget-Aware LLM Reasoning. ACL, 2025.
> - [5] Language Models (Mostly) Know What They Know. Anthropic, 2022.
> - [6] AbstentionBench: Reasoning LLMs Fail on Unanswerable Questions. NIPS, 2025.

---

> ### Author Response · Authors · 2025-11-25
> **Author response (2/3)**
>
> **Question 1**: Generalizability on coding tasks. The experiment mostly focuses on multiple choice questions or questions with numerical answers. Does it generalizes to coding questions? How would the methodology adapt to coding questions (and possibly more open-ended questions)?
>
> **Response**: Thank you for this insightful question! We fully agree with your concern regarding generalizability to coding tasks. We are sorry for not including such analysis in the original submission. However, our methods are designed to capture unsolvability signals from reasoning expressions and hidden states, **without relying on any task-specific heuristics**. Therefore, they should in principle generalize across different forms of reasoning tasks.
>
> To alleviate your concern, we conduct experiments on 1055 samples from LiveCodeBench [1]. The following table shows the accuracy of separating solvable and unsolvable coding questions based on capability boundaries revealed from reasoning expressions and hidden states.
>
> **Table.** Accuracy (%) in separating solvable and unsolvable questions via capability boundaries revealed by reasoning expressions (at 2% of the reasoning trace) and hidden states.
> | Model                 | ConfDiff (Reasoning Expressions) | ConfCurv (Reasoning Expressions) | LDA (Hidden States)    | LR (Hidden States)     |
> |-----------------------|----------|----------|--------|--------|
> | GPT-oss-20B           | 100      | 86.8     | 81.1  | 83.5  |
> | R1-Distill-Qwen3-8B   | 97.3     | 93.0     | 92.0  | 93.9  |
> | R1-Distill-Qwen3-32B  | 100      | 91.7     | 81.6  | 84.9  |
> | QwQ-32B               | 100      | 92.5     | 77.3  | 85.4  |
>
> We observe that **coding tasks also exhibit clear capability boundaries in both reasoning expressions and hidden states**. However, the accuracy from hidden states is lower than that from mathematical tasks. A plausible explanation is that the difficulty of coding problems is less explicitly signaled in the input itself. Unlike in math exams where humans can often immediately recognize “end-of-exam” questions that they are unlikely to solve, coding tasks usually require some degree of initial reasoning before the model can detect unsolvability. As a result, the unsolvability signal in the hidden states of the last input token is weaker, whereas the signal becomes more apparent in the early reasoning expressions.
>
> We will incorporate the above analysis into our revised submission.
>
> ---
>
> References
> - [1] LiveCodeBench: Holistic and Contamination Free Evaluation of Large Language Models for Code. Arxiv, 2024.

---

> ### Author Response · Authors · 2025-11-25
> **Author response (3/3)**
>
> **Question 2**: Ablation on the self-aware output prefix. Does changing the output prefix alter the result?
>
> **Response**: Thank you for this insightful comment! We fully agree with you that changing the output prefix will alter the results. Indeed, during our initial exploration, we experimented with several output prefixes and selected the one that achieved the largest reduction in token usage. We apologize for not describing this design choice in the submission. To address your concern, we conduct an ablation study comparing three output prefixes:
> - **Initial prefix (our early attempt):** Our goal is to prompt the model to acknowledge that it cannot solve an unsolvable question and then provide only a concise potential approach. However, we unexpectedly found that this cannot minimize token usage. The model’s instruction-following ability was insufficient for reliably producing only a concise approach, it often generated overly long responses. The initial prefix is:
> > \<think\>
> I think this question is beyond my capability boundary. I cannot fully solve it, but I can outline a concise potential approach.
> \</think\>
> This question is beyond my capability boundary, but I can outline a concise potential approach:
>
> - **Emphasis prefix (used in our submission):** Inspired by prior work [1,2], which shows that **the final sentence in the instruction strongly influences model behavior**, we appended `Step 1:` at the end of the instruction to explicitly trigger step-by-step approach. Also, we emphasize in the end of the `\<think\>...\</think\>` part that `I must give a concise outline to the user (less than 10 steps)!`. This combination elicits short, structured responses and lead to the largest reduction in token usage. This is the self-awareness prefix shown in our submission:
> > \<think\>
> I think this question is beyond my capability boundary. I cannot fully solve it, but I can outline a concise potential approach. I must give a concise outline to the user (less than 10 steps)!
> \</think\>
> This question is beyond my capability boundary, but I can outline a concise potential approach:
> Step 1:
>
> - **Token-budget prefix (following prior work [3]):** Following prior work [3] which compresses unnecessary reasoning by including a **reasonable token budget** in the prompt, we replace `less than 10 steps` with an explicit budget constraint `use less than 500 tokens`. The token-budget prefix is:
> > \<think\>
> I think this question is beyond my capability boundary. I cannot fully solve it, but I can outline a concise potential approach. I must give a concise outline to the user (use less than 500 tokens)!
> \</think\>
> This question is beyond my capability boundary, but I can outline a concise potential approach:
> Step 1:
>
> To avoid the confounding effect of incomplete answers and to more accurately assess changes in token usage, we set the sequence length limit to the maximum according to Huggingface model card.
>
> **Table.** Token usage under different output prefixes.
> | Output prefix    | Token |
> |------------------|--------|
> | No output prefix          | 5096   |
> | Initial          | 4423   |
> | **Emphasis**   | **2227**   |
> | Token-budget     | 2567   |
>
> The table above shows that **all three designs reduce token usage, with the Emphasis prefix achieving the largest reduction**. While it is possible that an even better prompt design exists, this does not affect our core conclusion: **setting a self-awareness output prefix enables the model to provide more efficient and more reliable responses on unsolvable questions**.
>
> ---
>
> References
> - [1] Understanding the Dark Side of LLMs' Intrinsic Self-Correction. ACL, 2025.
> - [2] Calibrate Before Use: Improving Few-shot Performance of Language Models. ICML, 2021.
> - [3] Token-Budget-Aware LLM Reasoning. ACL, 2025.

---

> > ### Comment · Reviewer_c4xf · 2025-11-25
> >
> > Thank you for the thorough response. All of my concerns have been addressed, and I’ve updated my score to reflect that.

---

> > > ### Author Response · Authors · 2025-11-25
> > >
> > > Thank you so much for your support!

---

### Official Review · Reviewer_keHX · 2025-11-02

**Soundness:** 2
**Presentation:** 2
**Contribution:** 3
**Rating:** 4
**Confidence:** 4

**Summary:**

The paper investigates whether open-source reasoning LRMs (Large Reasoning Models) possess self-awareness of their own capabilities - in particular, whether there’s some signal either in their reasoning trajectories or hidden states that would correlate with answer accuracy.

The authors find that by comparing the incidence rate of confident expressions  (e.g. “I am confident”, “this is solid”, “seems okay”, etc.) to those of unconfident expressions (e.g. “I’m stuck”, “not entirely confident”, “problematic”, etc.) in the models' reasoning trace, they’re able to predict with almost perfect accuracy whether the final answer is correct or incorrect on math benchmarks.

Motivated by their success of this approach even by analyzing only the beginning of the reasoning trace, the authors further investigate whether the LLM hidden states of the last input token are predictive of the correct answer. A logistic regression model is able to predict answer accuracy nearly perfectly across all the tested models.

Finally, the authors leverage both of these findings and propose two test-time monitoring strategies to either early-terminate the model’s reasoning based on the expression monitoring (black-box method), or have the model abstain from answering at all based on a classifier trained on the hidden state of the last input token (white-box method). Both strategies increase reliability and reduce token usage without sacrificing accuracy on the tested benchmarks (competition math, GSM8k, Humanity’s Last Exam).

**Strengths:**

- The paper is **well-structured, with all analyses and experiments clearly motivated and building upon each other naturally**. The initial study of reasoning expressions segues neatly into the analysis of input hidden states through the observation that capability boundaries are revealed already at an early stage of reasoning. Each of the two theoretical analyses are then subsequently utilized in practical test-time monitoring strategies.

- The paper **addresses a timely and important topic**: Do large reasoning models (LRMs) know their own capability boundaries?

- The **proposed test-time monitoring methods improve both reliability and token efficiency of open-source reasoning models** by having the model either outright abstain from providing a solution or cut short its reasoning process when the question is deemed outside its capability boundary. For example, the biggest reasoning model the authors studied, QwQ-32B, showed perfect abstention rate with “expression monitoring” and ~50% reduction in token usage on unsolvable problems, without any loss in accuracy (Table 3, last row).

**Weaknesses:**

- Throughout, the paper **lacks sufficient level of detail about the experimental setup**
  - The fundamental section of the paper is section 4, wherein the authors analyse the correlation between answer accuracy and incidence rate of various “reasoning expressions” in the reasoning trace. The paper doesn’t elaborate on how the specific expressions (provided in Appendix C) were chosen, nor provides any quantitative analysis of relative importance of individual reasoning expressions. Finally, the exact method by which the reasoning expressions are used to estimate confidence levels is left ambiguous (the authors do mention “manual summarization and LLMs verification”, but no further detail is provided).
  - Section 4 figure 4 - what sampling settings were used, what max sequence length for the output?
  - Section 5 figure 5 - how is  “close / far from the decision boundary” quantified?
  - In Section 6.1, the authors define the Hard Abstention (HA) metric, which “evaluates whether LRMs can explicitly abstain on unsolvable problems instead of producing incorrect answers”. However, the paper doesn’t specify how abstention is detected.

- In Section 6.1, the proposed test-time monitoring strategies are evaluated against a single, relatively weak baseline—BoostAbstention, a system prompt encouraging the model to abstain on difficult problems. This baseline yields a 0% Hard Abstention (HA) rate across all settings, providing little insight into how the proposed methods compare to realistic alternatives.
**The paper would benefit from the inclusion of stronger baselines to help better contextualize the proposed improvements**, such as model-as-judge or uncertainty-estimation approaches (e.g., Wen et al., 2025 for an overview).

- In section 6, table 3 reports that the 2k and 4k sequence length limits are exceeded by all analysed models in almost 100% of incorrect cases. If I’m reading this right, then the vanilla QwQ-32B produces no outright incorrect completions - in every case it didn’t arrive at a correct answer, it simply ran out of context. This suggests that the proposed monitoring strategies might primarily identify context-limited or under-budgeted reasoning chains rather than truly unsolvable problems - **the paper would benefit from extending the analyses to longer sequence lengths.**

- The evaluation results in Section 6 aggregate across six benchmarks, but Appendix A shows that roughly 96% of all evaluation problems (~3k / 3.12k) come from GSM8K and Humanity’s Last Exam (HLE)—two datasets that sit at opposite ends of the accuracy spectrum (≈ 90% vs < 10%).
Consequently, many reported metrics—particularly the Hard Abstention (HA) scores in Table 3—likely conflate benchmark difficulty with model “self-awareness.” In effect, the monitoring strategies may simply learn to distinguish which dataset a question originates from, rather than whether a question is intrinsically within or beyond the model’s capability boundary.
This same imbalance likely inflates the near-perfect linear-probe accuracies reported in Section 5, since solvable (GSM8K) and unsolvable (HLE) examples are trivially separable.
**The paper would benefit from reporting per-benchmark metrics (HA, token usage, overflow, etc.) to disambiguate the effects of benchmark distribution from genuine boundary detection.**

- In all experiments in the paper (e.g. section 5), the authors train and test on different subsets of the same datasets. **The paper could benefit from cross-dataset generalization tests (e.g., train on GSM8K, test on AIME).**

**Questions:**

See “weaknesses” section

---

> ### Author Response · Authors · 2025-11-25
> **Author response (1/6)**
>
> Dear Reviewer keHX, thank you very much for your careful review of our paper and thoughtful comments. We are encouraged by your positive comments on **good and natural structure, timely and important topic, and reliable and token efficient test-time monitoring methods** of our paper. We hope the following responses can help clarify potential misunderstandings and alleviate your concerns.
>
> ---
>
> **Weakness 1**: The paper lacks sufficient level of detail about the experimental setup.
>
> **Response**: Thank you for this comment! We are sorry for not providing sufficient experimental details in our initial submission. Below, we try to alleviate the specific issues you raised regarding the experimental details.
>
> **Issue 1: Regarding reasoning expressions.**
>
> **Issue 1.1: How the specific expressions were chosen? Quantitative analysis of relative importance?**
>
> We provide a more detailed description of the pipeline for identifying reasoning expressions of confidence:
> - 1. **Identify all anthropomorphic expressions from reasoning.** Prior work [1,2] show that the anthropomorphic expressions (e.g., "wait") are crucial for LRM reasoning performance, in contrast to purely neutral mathematical steps (e.g., "11 + 5 = 16"). Following this insight, we manually curate a set of anthropomorphic expressions that LRMs commonly use. And we further verify with GPT-5 that these expressions indeed convey anthropomorphic tone rather than purely propositional content.
> - 2. **Select expressions whose occurrence frequency is most predictive of answer correctness.** For each question, we count the occurrences of all extracted expressions within reasoning. Using the final answer correctness as the label and the occurrence counts for each expressions as features, we apply feature-importance analysis methods [3], including SHAP and Permutation Importance, to identify the subset of expressions most informative for predicting whether the model’s final answer is correct. As shown in the following table, we select expressions such as “wait”, “yes”, “exactly”, and “missed” which rank among the most predictive, whereas others (e.g., “so”) exhibit negligible predictive value and are therefore excluded.
> - 3. **Divide the selected expressions into confidence-related and uncertainty-related categories.** This step is grounded in the connection between epistemic markers and expressed uncertainty. Prior work [4,5] show that epistemic markers reflect LLMs’ internal confidence, analogous to how humans use expressions such as “I am not sure” or “it is unlikely that” to convey uncertainty. Following this insight, we manually classify each selected expression as reflecting confidence or uncertainty, and we further validate this categorization using annotations generated by GPT-5.
>
> **Table.** Feature importance score of expressions.
> | Expression | SHAP ↑ | Permutation Importance ↑ | Combined Score ↑ |
> |-----------|------|--------------------------|----------------|
> | wait      | 1.00 | 1.00                     | 2.00           |
> | yes       | 0.33 | 0.73                     | 1.06           |
> | exactly   | 0.26 | 0.73                     | 0.99           |
> | missed    | 0.10 | 0.86                     | 0.97           |
> | ...       |      |                          |                |
> | so        | 0.01 | 0.00                     | 0.01           |
>
>
> **Issue 1.2: How to use the expressions to estimate confidence level?**
>
> We estimate the model’s confidence level from the occurrences of confidence-related and uncertainty-related expressions throughout the reasoning process, following prior work [5] which measures LLM uncertainty by counting the frequency of epistemic markers. In Section 3 of our submission, we use the category of expressions that appears more frequently near the end of the reasoning trace to determine whether the model is more confident or more uncertain. Building on this idea, Section 4 analyzes the trajectory of confidence and uncertainty signals over the entire reasoning sequence, captured by ConfDiff and ConfCurv (equation 2 and 3 in our submission), to assess whether the model is more confident or more uncertain.
>
> ---
>
> References
> - [1] DeepSeek-R1: Incentivizing Reasoning Capability in LLMs via Reinforcement Learning. Nature, 2025.
> - [2] Demystifying Reasoning Dynamics with Mutual Information: Thinking Tokens are Information Peaks in LLM Reasoning. NIPS, 2025.
> - [3] Feature Selection Strategies: a Comparative Analysis of SHAP-value and Importance-based Methods. Journal of Big Data, 2024.
> - [4] Can Large Language Models Faithfully Express Their Intrinsic Uncertainty in Words? EMNLP, 2024.
> - [5] Revisiting Epistemic Markers in Confidence Estimation: Can Markers Accurately Reflect Large Language Models' Uncertainty? ACL, 2025.

---

> ### Author Response · Authors · 2025-11-25
> **Author response (2/6)**
>
> **Issue 2: Section 4 figure 4 - sampling settings and max sequence length?**
>
> For the sampling settings, we use greedy decoding to ensure reproducibility of results. For the maximum sequence length, we follow the specifications in the HuggingFace model cards: 128k for GPT-oss-20B, 64k for R1-Distill-Qwen3-8B, and 32k for both R1-Distill-Qwen3-32B and QwQ-32B.
>
> **Issue 3: Section 5 figure 5 - how is “close / far from the decision boundary” quantified?**
>
> For the hidden states corresponding to each solvable question, we compute the difference between the probabilities assigned by the LDA linear classifier to the solvable and unsolvable classes. This follows prior work [6] that uses probability differences as a proxy for distance to the decision boundary in binary classification. We then define the bottom 1% of data with the smallest differences as close to the decision boundary, and the remaining points as far from the decision boundary.
>
> **Issue 4: How abstention is detected?**
>
> In Section 4 and Section 5 of our submission, we evaluate the accuracy of separating solvable and unsolvable questions (Figure 4 and Table 2) using capability boundaries revealed by reasoning expressions (ConfDiff) and hidden states (linear classifiers LDA). Based on these signals, hard abstention (HA) is defined as the recall of unsolvable questions detected by ConfDiff and LDA, which correspond to HA of Monitor$\_{\text{express}}$ and Monitor$\_{\text{hidden}}$ in Table 3. Once an unsolvable question is detected, our monitoring strategies will make the model to explicitly acknowledge “This question is beyond my capability boundary...” and to provide a concise outline of possible solution approaches, rather than producing a wrong answer (as shown in Figure 6).
>
> We will incorporate the above clarifications into the revised version of our submission.
>
> ---
>
> References
> - [6] Active Learning for Semantic Segmentation with Multi-class Label Uncertainty. NIPS, 2023.

---

> ### Author Response · Authors · 2025-11-25
> **Author response (3/6)**
>
> **Weakness 2**: The paper would benefit from the inclusion of stronger baselines to help better contextualize the proposed improvements.
>
> **Response**: Thank you for this comment! As suggested, we provide comparisons to model-as-judge approach [1] and uncertainty-estimation approach [2,3]. However, our work study **abstention on unsolvable questions**, and therefore these methods cannot directly serve as a baseline for our setting. Consequently, we first clarify how we adapt these two methods to fit our task.
>
> **Model-as-judge:** For each question, we ask the model to evaluate whether its own predicted answer is correct using the following prompt template:
> > Question: {question}
> Proposed Answer: {answer}
> Is the proposed answer:
> A True
> B False
> The proposed answer is:
>
> This follows prior work [1] which study whether language models can evaluate the validity of their own claims by asking models to first propose answers, and then to evaluate that their answers are correct. If the model judges its own incorrect answer to be incorrect, we count this as a hard abstention (HA). Conversely, if the model judges its correct answer to be incorrect, this reduces the overall task accuracy (Acc). Formally,
>
> $
> \text{Acc} = \mathbb{P} \left(
>     \text{Model judges the answer to be correct} \land
>     \text{Answer is correct}
> \right)
> $
>
> $
> \text{HA} = \mathbb{P} \left(
>     \text{Model judges the answer to be incorrect}
>     \mid
>     \text{Answer is incorrect}
> \right)
> $
>
> **Uncertainty-estimation:** For each question, we compute the semantic entropy of the model’s generated answer, where a higher entropy indicates greater uncertainty. This follows prior work [2,3], which introduces semantic entropy as an uncertainty measure that captures linguistic invariances induced by shared meanings. When the entropy exceeds a certain threshold, the model is considered uncertain about its answer. Similar as before, we define:
>
> $
> \text{Acc} = \mathbb{P} \left(
>     \text{Semantic entropy of the answer < threshold} \land
>     \text{Answer is correct}
> \right)
> $
>
> $
> \text{HA} = \mathbb{P} \left(
>     \text{Semantic entropy of the answer >= threshold}
>     \mid
>     \text{Answer is incorrect}
> \right)
> $
>
> **Results:** We provide the results of these two baselines alongside our Monitor$\_{\text{express}}$ and Monitor$\_{\text{hidden}}$ in the following table. **Although both baselines provide models with a certain degree of hard abstention capability, they also incorrectly classify many solvable questions as unsolvable, which significantly reduces overall accuracy.** In contrast, our approaches enable LRMs to reliably identify unsolvable questions without compromising their accuracy on solvable ones.
>
> **Table.** Accuracy (Acc) and Hard Abstention (HA) of two baselines alongside our monitoring strategies.
> | Method                 | Acc ↑  | HA ↑|
> |------------------------|--------|--------|
> | GPT-oss-20B            | 74.6%  | 0      |
> | + Model-as-judge       | 32.1%  | 46.0%  |
> | + Uncertainty-estimation | 50.0%  | 29.0%     |
> | + Monitor$\_{\text{express}}$    | 74.6%  | 100%   |
> | + Monitor$\_{\text{hidden}}$     | 73.7%  | 98.9%  |
>
> ---
>
> References
> - [1] Language Models (Mostly) Know What They Know. Anthropic, 2022.
> - [2] Semantic Uncertainty: Linguistic Invariances for Uncertainty Estimation in Natural Language Generation. ICLR, 2023.
> - [3] Detecting Hallucinations in Large Language Models Using Semantic Entropy. Nature, 2024.

---

> ### Author Response · Authors · 2025-11-25
> **Author response (4/6)**
>
> **Weakness 3**: The paper would benefit from extending the analyses to longer sequence lengths.
>
> **Response**: Thank you for this insightful comment! We fully understand your concern that models with 2k or 4k sequence length limit may not be able to produce complete answers. However, there might be a potential misunderstanding.
>
> **Our proposed monitoring strategies are not constrained by sequence length limit**:
> - Monitor$\_{\text{express}}$ continuously tracks the densities of confident and uncertain expressions throughout the reasoning process and can detect unsolvable questions **at an early stage** (Figure 4), well before the model reaches the end of reasoning.
> - Monitor$\_{\text{hidden}}$ identifies unsolvable questions directly from the hidden states of last input token, i.e., **even before the model begins generating its reasoning**.
>
> Because our monitoring strategies operate early in the reasoning trajectory, **we intentionally used relatively short sequence length limits (e.g., 2k or 4k) in the initial submission to demonstrate their early detection capabilities**.
>
> To further alleviate your concern, we provide the results (in the following table) **under extended sequence length limits, set to the maximum** allowed by each model according to its HuggingFace model card (128k for GPT-oss-20B, 64k for R1-Distill-Qwen3-8B, and 32k for both R1-Distill-Qwen3-32B and QwQ-32B). Under these settings, the models always produce complete answers (Overflow = 0), and **our monitoring strategies still reduce token usage significantly as expected** (Acc and HA are the same as under 2k or 4k sequence length limit).
>
> **Table.** Results under maximum sequence length.
> | Method                     | Sequence length | Token ↓ | Overflow ↓ |
> |----------------------------|------------------|-------|----------|
> | GPT-oss-20B                | 128k             | 6023  | 0        |
> | + Monitor$\_{\text{express}}$        | 128k             | 1897  | 0        |
> | + Monitor$\_{\text{hidden}}$         | 128k             | 3826  | 0        |
> | R1-Distill-Qwen3-8B        | 64k              | 5096  | 0        |
> | + Monitor$\_{\text{express}}$        | 64k              | 2834  | 0        |
> | + Monitor$\_{\text{hidden}}$         | 64k              | 2227  | 0        |
> | R1-Distill-Qwen3-32B       | 32k              | 5910  | 0        |
> | + Monitor$\_{\text{express}}$        | 32k              | 1043  | 0        |
> | + Monitor$\_{\text{hidden}}$         | 32k              | 1246  | 0        |
> | QwQ-32B                    | 32k              | 5062  | 0        |
> | + Monitor$\_{\text{express}}$        | 32k              | 2002  | 0        |
> | + Monitor$\_{\text{hidden}}$         | 32k              | 274   | 0        |

---

> ### Author Response · Authors · 2025-11-25
> **Author response (5/6)**
>
> **Weakness 4**: In effect, the monitoring strategies may simply learn to distinguish which dataset a question originates from, rather than whether a question is intrinsically within or beyond the model’s capability boundary ... The paper would benefit from reporting per-benchmark metrics (HA, token usage, overflow, etc.) to disambiguate the effects of benchmark distribution from genuine boundary detection.
>
> **Response**: Thank you for this insightful comment! We fully agree with you that, because 96% questions come from GSM8K and HLE, the HA metric might conflate benchmark difficulty with model "self-awareness". However, there might be a potential misunderstanding: **the ability to recognize whether a question comes from an easy or hard question distribution is not in conflict with a model’s self-awareness of capability boundary**.
>
> To illustrate, consider humans with self-awareness of capability boundary. In an exam with limited time and many hard problems, skilled test-takers are able to quickly identify questions beyond their capability and avoid wasting time on them. They may recognize that the question resembles ones they consistently failed to solve during preparation (analogous to our Monitor$\_{\text{hidden}}$ strategy), or they may attempt a few initial reasoning steps and realize they lack confidence to proceed (analogous to our Monitor$\_{\text{express}}$ strategy).
>
> Still, we find your suggestion on "reporting per-benchmark metrics" extremely valuable. We provide in the following table the accuracy of separating unsolvable questions from solvable ones **within each benchmark using LDA and LR classifiers trained and tested on splits from the same dataset**. Specifically, for both HLE and GSM8K, we randomly split each dataset into an 80% training set and a 20% test set (with balanced solvable/unsolvable labels), train the linear classifier on the training set, and evaluate its classification accuracy on the test set.
>
> **Table.** Accuracy (%) of separating solvable and unsolvable questions in hidden states using linear classifiers trained and tested on subsets split within the same benchmark.
> | Dataset | Model                  | LDA  | LR   |
> |---------|-------------------------|------|------|
> | HLE     | GPT-oss-20B            | 89.3 | 90.3 |
> | HLE     | R1-Distill-Qwen3-8B    | 94.9 | 96.4 |
> | HLE     | R1-Distill-Qwen3-32B   | 96.4 | 96.4 |
> | HLE     | QwQ-32B                | 93.4 | 96.4 |
> | GSM8k   | GPT-oss-20B            | 80.3 | 83.3 |
> | GSM8k   | R1-Distill-Qwen3-8B    | 70.1 | 71.6 |
> | GSM8k   | R1-Distill-Qwen3-32B   | 85.2 | 89.0 |
> | GSM8k   | QwQ-32B                | 87.9 | 87.1 |
>
> The results show that **linear classifiers can still separate solvable and unsolvable questions from hidden states, although with reduced accuracy compared to aggregated benchmarks setting (Table 2)**. This drop is small on HLE but larger on GSM8K. The reason is that **HLE contains a larger proportion of unsolvable questions, enabling the linear classifiers to better learn the hidden state information associated with unsolvability**. Consequently, these unsolvable questions in HLE can be effectively leveraged to develop models' Hard Abstention capability.

---

> ### Author Response · Authors · 2025-11-25
> **Author response (6/6)**
>
> **Weakness 5**: The paper could benefit from cross-dataset generalization tests (e.g., train on GSM8K, test on AIME).
>
> **Response**: Thank you for this insightful comment! In the above analysis, we think the unsolvable questions in HLE can be effectively leveraged to develop models' Hard Abstention capability. Therefore, we train linear classifiers on HLE and evaluate models' Hard Abstention capability on GSM8k and AIME to test whether this capability generalizes across benchmarks. As shown in the following table, **linear classifiers trained on HLE are able to identify nearly all unsolvable questions in GSM8K and AIME**, achieving HA close to 100%.
>
> **Table.** Hard Abstention (%) of linear classifier trained on HLE and test on GSM8k and AIME.
> | Model                 | GSM8k | AIME |
> |-----------------------|--------|------|
> | GPT-oss-20B           | 0      | 0    |
> | + Monitor$\_{\text{hidden}}$    | 100    | 100  |
> | R1-Distill-Qwen3-8B   | 0      | 0    |
> | + Monitor$\_{\text{hidden}}$    | 100    | 100  |
> | R1-Distill-Qwen3-32B  | 0      | 0    |
> | + Monitor$\_{\text{hidden}}$    | 90.5   | 94.7 |
> | QwQ-32B               | 0      | 0    |
> | + Monitor$\_{\text{hidden}}$    | 98.9   | 100  |
>
> In contrast, training on GSM8K is insufficient for developing Hard Abstention capability, because GSM8K contains very few unsolvable questions. The linear classifiers cannot reliably learn the hidden state patterns associated with unsolvability.

---

> ### Author Response · Authors · 2025-11-28
> **Thanks to Reviewer keHX**
>
> Please allow us to thank you again for reviewing our paper and the valuable feedback, and in particular for recognizing the strengths of our paper in terms of **good and natural structure, timely and important topic, and reliable and token efficient test-time monitoring methods**.
>
> Kindly let us know if our response and the new experiments have properly addressed your concerns. We are more than happy to answer any additional questions during the rebuttal period. Your feedback will be greatly appreciated.

---

### Author Response · Authors · 2025-12-01
**Summary of Rebuttal (1/3)**

Dear Area Chair,

Greetings and wish you all the best. We deeply and sincerely appreciate your time devoted to reviewing our manuscript.

We fully respect the new ICLR policy and confirm that we never use the bug to manipulate the scores. However, we sincerely hope that the reviewers’ positive responses to our rebuttal (**scores updated from 66442 to 86644**: Reviewer c4xf raised the score from 6 → 8; Reviewer DRCG raised the score from 2 → 6) will not be overshadowed by the score-reverting procedure.

Below, we provide a summary to facilitate your quick tracking of the review process.

---

# Paper Summary

Our work stems from a common phenomenon: **When faced with hard questions, LRMs often engage in unproductive reasoning, producing wrong answers while wasting substantial computation**.

This phenomenon highlights a fundamental issue: **Current answering paradigms of LRMs overlook the relationship between questions and their capability boundaries**.

Thus, we investigate the question: **Do LRMs know their capability boundaries?**

This research question has been **rarely examined and offers a complementary perspective to the widely studied domain of efficient reasoning**: prior works [1-6] focus on mitigating overthinking on solvable questions, whereas we focus on abstaining from unsolvable questions and producing concise solution outlines instead.

We identify boundary-awareness signal in reasoning expressions (for black-box models) and hidden states of last input token (for white-box models). Building on these findings, we design two test-time monitoring strategies to improve answer efficiency and reliability without sacrifying accuracy.

---

# Positive Comments

**Important research problem whose significance is widely recognized**:
- Reviewer keHX: "The paper addresses a timely and important topic."
- Reviewer 93Mj: "this paper addresses an important research problem whose significance is widely recognized."

**Novel and interesting analysis and methodology**:
- Reviewer c4xf: "The analysis and methodology is novel and interesting."
- Reviewer haJD: "Evidence comes from both black-box signals (confidence/uncertainty trajectories) and white-box signals (last-token hidden states), which makes the story feel cohesive."
- Reviewer DRCG: "The perspective of understaing model capacity through certainty trajectory's concavity is novel"; "The authors design comprehensive experiments on multiple models and benchmarks, supporting a universality of their findings."


**Reliable and token efficient test-time monitoring methods**:
- Reviewer keHX: "The proposed test-time monitoring methods improve both reliability and token efficiency of open-source reasoning models by having the model either outright abstain from providing a solution or cut short its reasoning process when the question is deemed outside its capability boundary."
- Reviewer c4xf: "The methodology proposed is sound, intuitive, effective in practice, and backed by a lot of experiments and ablations to show its effectiveness."
- Reviewer haJD: "Reported token savings are large without hurting accuracy, suggesting real efficiency wins for math-style workloads."


**Good structure, natural logic flow, clear presentation**:
- Reviewer keHX: "The paper is well-structured, with all analyses and experiments clearly motivated and building upon each other naturally."
- Reviewer c4xf: "The evidence is presented very cleanly, and the logic flows very naturally."
- Reviewer 93Mj: "Its biggest strength in my opinion is the clarity of presentation along with supporting results."
- Reviewer haJD: "The paper frames 'capability boundaries' clearly and turns that idea into two simple, test-time strategies that are easy to implement."

---

> ### Author Response · Authors · 2025-12-01
> **Summary of Rebuttal (2/3)**
>
> # Summary of Reviewer Interactions and Our Improvements
>
> To alleviate reviewers' concerns, we conduct **6 additional experiments** and provide expanded analyses. We have also submitted the revised manuscript, in which **all modifications are highlighted in blue**.
>
> **Reviewer keHX (No Reply Yet)**
>
> - **Lack of detail about the experimental setup**. We provide detailed responses to all points raised (line 316-319, 826-829), and we add a more detailed pipeline for identifying reasoning expressions (Section 4.1 and Appendix C).
> - **Inclusion of stronger baseline**. We incorporate comparison to stronger baselines (line 424-429, Appendix B), showing that our approaches enable LRMs to reliably identify unsolvable questions without compromising their accuracy on solvable ones.
> - **Extending the analyses to longer sequence lengths**. We provide the results under extended sequence length limits (line 502-503, Appendix I), showing that our monitoring strategies still reduce token usage significantly as expected.
> - **Per-benchmark metrics**. We include results separating solvable vs. unsolvable questions within each benchmark (Appendix F), confirming that the two types remain distinguishable through hidden states.
> - **Cross-dataset generalization tests**. We provide the results of training on HLE and testing on GSM8k and AIME (Appendix F), showing that unsolvable questions in HLE can be effectively leveraged to develop models' Hard Abstention capability.
>
> **Reviewer c4xf (Score raised: 6 → 8)**
>
> Reviewer c4xf states that all the following concerns have been addressed and has raised the score from 6 → 8.
> - **Expanding related work**. We clarify the distinction between our work and prior studies [1-5] in both research question and methodology (line 125-134).
> - **Generalizability on coding tasks**. We conduct experiments on LiveCodeBench [7] (line 346-348, Appendix G), showing that coding tasks also exhibit clear capability boundaries in both reasoning expressions and hidden states.
> - **Ablation on the self-aware output prefix**. We conduct an ablation study comparing three output prefixes (line 395, Appendix H), showing that all three designs reduce token usage, with the one used in our submission achieving the largest reduction.

---

> ### Author Response · Authors · 2025-12-01
> **Summary of Rebuttal (3/3)**
>
> **Reviewer 93Mj (No Reply Yet)**
>
> - **Contributions**. We clarify the distinction between our work and prior studies [1-6]: they focus on mitigating overthinking on solvable questions, whereas we focus on abstaining from unsolvable questions and producing concise solution outlines instead.
> - **Focus on scenarios where LLM has reasonable accuracy**. We clarify a possible misunderstanding: our experiments already focus on such scenarios. They are conducted on a dataset mixing six benchmarks covering a wide range of difficulty levels, and we provide an explanation of the rationale behind this design (lines 296–298, Appendix F).
>
> **Reviewer haJD (No Reply Yet)**
>
> - **Hidden-state separability within a single dataset**. We include results separating solvable vs. unsolvable questions within a single dataset (Appendix F), confirming that the two types remain distinguishable through hidden states.
> - **Sensitivity of black-box signals**. We clarify that the reasoning expressions (i.e., black-box signals) are individually extracted for each LRM and for each task, and we provide a more detailed pipeline for identifying these expressions (Section 4.1 and Appendix C).
> - **Stronger uncertainty/abstention baselines**. We incorporate comparison to stronger baselines (line 424-429, Appendix B), showing that our approaches enable LRMs to reliably identify unsolvable questions without compromising their accuracy on solvable ones.
>
> **Reviewer DRCG (Score raised: 2 → 6)**
>
> Reviewer DRCG states that some of the following concerns have been addressed and has raised the score from 2 → 6.
> - **Lack of some important details**. We provide detailed responses to all points raised (line 231-234, 244-245, 296-298), and we add a more detailed pipeline for identifying reasoning expressions (Section 4.1 and Appendix C).
> - **Results without any overflow constraint**. We provide the results without overflow constraint (line 502-503, Appendix I), showing that our monitoring strategies still reduce token usage significantly as expected.
> - **Comparing with some other confidence based early existing methods**. We compare with a recent approach for improving reasoning efficiency [8], showing its failure on detecting unsolvable quesitions.
>
> ---
>
> We sincerely believe our responses and revisions have addressed most reviewers' concerns and help us better deliver this work to the ICLR community.
>
> Thank you again for your efforts on this paper and for your contributions to the community！
>
> ---
>
> References
> - [1] ThinkPrune: Pruning Long Chain-of-Thought of LLMs via Reinforcement Learning. Arxiv, 2025.
> - [2] Deep Think with Confidence. Arxiv, 2025.
> - [3] Efficiently Scaling LLM Reasoning with Certaindex. NIPS, 2025.
> - [4] Token-Budget-Aware LLM Reasoning. ACL, 2025.
> - [5] Language Models (Mostly) Know What They Know. Anthropic, 2022.
> - [6] Reasoning Models Know When They're Right: Probing Hidden States for Self-Verification. COLM, 2025.
> - [7] LiveCodeBench: Holistic and Contamination Free Evaluation of Large Language Models for Code. Arxiv, 2024.
> - [8] Dynamic Early Exit in Reasoning Models. Arxiv, 2025.

---

### Meta-Review · Area_Chair_JK3n · 2026-01-06

**Summary:**

1) Three of the reviewer (keHX, DRCG, haJD) were concerned by the lack of clarity in the description of the experimental setup.

2) Some Reviewers (DRCG,keHX) criticized the breadth of the experimental evaluation.

3) Generality and deployability: the applicability of the proposed method beyond math tasks seems to have been an issue for Reviewer c4xf and Reviewer haJD.

4) Multiple reviewers had an issue with the term "self-awareness" and/or "capability boundary" and how they are probed in the experimental evaluation. Here are the respective quotes from the review:

- Reviewer kehX: Consequently, many reported metrics—particularly the Hard Abstention (HA) scores in Table 3—likely conflate benchmark difficulty with model ‘self-awareness.’ In effect, the monitoring strategies may simply learn to distinguish which dataset a question originates from, rather than whether a question is intrinsically within or beyond the model’s capability boundary.

- Reviewer haJD: The striking linear separability in hidden states may reflect dataset or difficulty splits rather than genuine boundary awareness, so the claim risks a distribution confound.

- Reviewer DRCG: What is the definition of capability boundary?

The criticism that is then formulated by the reviewer is that it is not clear whether the experiments measure the "capability boundary" of a model or whether the experiments measure the hardness of a dataset.

**Reviewer Concerns:**

The rebuttal was quite extensive and the authors have added written a detailed rebuttal. Furthermore, they have considerably amended the manuscript. This allowed them to address the most critical points raised by the reviewers.

1) The authors substantially expanded methodological details, e.g. clarified the expression-selection pipeline (feature-importance–based, per-model/task), defined reasoning stages and online monitoring, specified decoding/sequence lengths, and clarified HA/abstention detection. Reviewers has DRCG engaged in the rebuttal process and acknowledged these clarifications have addressed their respective concern.

2) The authors have added comparisons to experimental results, which show that the additional baselines requested by the reviewers are outperformed by their proposed method. Reviewer DRCG indicated this concern was addressed.

3) The authors added coding-task experiments (LiveCodeBench), conducted prefix ablations, and clarified that black-box monitoring is intended as the deployable alternative when hidden states are unavailable. Reviewer c4xf confirmed these changes addressed their concerns.

4) The authors added per-benchmark, within-dataset, and cross-dataset experiments, and provided a conceptual argument (via a human-exam analogy) that recognizing difficulty distributions is consistent with capability awareness. While the additional experiments strengthen the empirical case, the concern is not fully resolved, as it still hinges on interpretation. The core issue is the framing and strength of the claim: whether difficulty/distribution recognition should be interpreted as “capability boundary awareness”. This is the primary remaining substantive concern. One possible way to actually distinguish these two concepts would be to establish that there is no total order of benchmarks in terms of hardness. Then one would need two models. Each one performs poorly on one problem and well on the other (each model has a "favored" dataset different from the other models dataset). These models would then need to detect whether they can solve a problem or not and get high accuracy in this prediction, independently of the dataset. I think all of these issues come down to not having formal definitions of "capability boundary" contrasted to the hardness of a dataset.

**Reviewer Scores:**

Two reviewer have indicated in comment that they would increase their scores.
Reviewer c4xf: 6->8
Reviewer DRCG: 2->4/6 (the authors claim 6 but I cannot verify this)

---

### Decision · Program_Chairs · 2026-01-26

Reject